# Mediator MED23 regulates inflammatory responses and liver fibrosis

**Zhichao Wang**[1☯]**, Dan Cao**[2☯]**, Chonghui Li**[2]**, Lihua Min**[2]**, Gang Wang**[1]*

**1** State Key Laboratory of Genetic Engineering, School of Life Sciences and Zhongshan Hospital, Fudan University, Shanghai, China, **2** State Key Laboratory of Cell Biology, CAS Center for Excellence in Molecular Cell Science, Shanghai Institute of Biochemistry and Cell Biology, Chinese Academy of Sciences, University of Chinese Academy of Sciences, Shanghai, China

☯ These authors contributed equally to this work.
* gwang_fd@fudan.edu.cn

**Data Availability Statement:** All relevant data except raw RNA sequencing data are within the paper and its Supporting Information files. Raw RNA sequencing data were deposited in the National Center for Biotechnology Information

## Abstract

Liver fibrosis, often associated with cirrhosis and hepatocellular carcinomas, is characterized by hepatic damage, an inflammatory response, and hepatic stellate cell (HSC) activation, although the underlying mechanisms are largely unknown. Here, we show that the transcriptional Mediator complex subunit 23 (MED23) participates in the development of experimental liver fibrosis. Compared with their control littermates, mice with hepatic *Med23* deletion exhibited aggravated carbon tetrachloride ($CCl_4$)-induced liver fibrosis, with enhanced chemokine production and inflammatory infiltration as well as increased hepatocyte regeneration. Mechanistically, the orphan nuclear receptor RAR-related orphan receptor alpha (RORα) activates the expression of the liver fibrosis-related chemokines C-C motif chemokine ligand 5 (CCL5) and C-X-C motif chemokine ligand 10 (CXCL10), which is suppressed by the Mediator subunit MED23. We further found that the inhibition of *Ccl5* and *Cxcl10* expression by MED23 likely occurs because of G9a (also known as euchromatic histone-lysine N-methyltransferase 2 [EHMT2])-mediated H3K9 dimethylation of the target promoters. Collectively, these findings reveal hepatic MED23 as a key modulator of chemokine production and inflammatory responses and define the MED23-CCL5/CXCL10 axis as a potential target for clinical intervention in liver fibrosis.

## Introduction

Liver fibrosis is a major chronic liver disease that can progress to more severe liver cirrhosis and eventually cause hepatocellular carcinoma, accompanied by significant mortality [1, 2]. Sufficient evidence supports the hypothesis that liver fibrosis is the consequence of the wound-healing response, which maintains the original architecture to accommodate the compensatory proliferation of hepatocytes [3]. It is worth noting that despite the annual increases in the prevalence and risk of liver fibrosis, especially in Asian countries, there is no proven effective treatment strategy to date [4]. Thus, further understanding of the molecular pathophysiology of liver fibrosis and development of mechanism-based therapeutics are urgently needed.

(NCBI), Gene Expression Omnibus database under accession number GSE137457.

**Funding:** This work was supported in part by grants from the National Natural Science Foundation Grant of China (31671543) to GW and the Ministry of Science and Technology of China (2017YFA0102700) to GW.The funders had no role in study design, data collection and analysis, decision to publish, or preparation of the manuscript.

**Competing interests:** The authors have declared that no competing interests exist.

**Abbreviations:** AML12, alpha mouse liver 12; ALT, alanine aminotransferase; α-SMA, alpha-smooth muscle actin; AST, aspartate aminotransferase; CCL, C-C motif chemokine ligand; CCl$_4$, carbon tetrachloride; CCR, C-C motif chemokine receptor; CEBPB, CCAAT enhancer binding protein beta; ChIP, chromatin immunoprecipitation; cIAP, cellular inhibitor of apoptosis protein; Col, collagen; Col1a1, collagen type I alpha 1 chain; Col3a1, collagen type III alpha 1 chain; CRISPR/Cas9, clustered regularly interspaced short palindromic repeats/CRISPR-associated protein 9; CXCL, C-X-C motif chemokine ligand; Cxcr2, C-X-C motif chemokine receptor 2; ECM, extracellular matrix; EHMT2, euchromatic histone-lysine N-methyltransferase 2; GAPDH, glyceraldehyde 3-phosphate dehydrogenase; GSEA, gene set enrichment analysis; HCC, hepatocellular carcinoma; HDAC, histone deacetylase; HE, hematoxylin–eosin; HFD, high-fat diet; Hgf, hepatocyte growth factor; HSC, hepatic stellate cell; HSP70, hot shock protein 70; Il-6, interleukin 6; IPA, Ingenuity Pathway Analysis; MCD, methionine and choline-deficient; MED23, Mediator complex subunit 23; med23$^{\Delta li}$, liver-specific knockout of *Med23*; med23$^{f/f}$, *med23*-floxed; MMP, matrix metalloproteinase; NASH, nonalcoholic steatohepatitis; NF-κB, nuclear factor kappa-light-chain-enhancer of activated B cells; NK, natural killer; PCNA, proliferating cell nuclear antigen; Pdgfβ, platelet-derived growth factor beta; Pdgfrβ, platelet-derived growth factor receptor beta; Pol II, RNA polymerase II; PPARα, peroxisome proliferator–activated receptor alpha; qRT-PCR, quantitative real-time PCR; RANTES, regulated upon activation normal T cell expressed and secreted factor; REST, RE1 silencing TF; γ-H2AX, phosphorylation of the histone variant H2AX; RNAi, RNA interference; RORα, RAR-related orphan receptor alpha; RORE, RORα response element; sg, staggerer; siRNA, small interfering RNA; TF, transcription factor; Tgfβ1, transforming growth factor beta 1; Tgfβr1, transforming growth factor beta receptor 1; TIMP, tissue inhibitor of

Several studies to date have demonstrated that hepatic inflammation plays an important role in the underlying pathogenesis of liver fibrosis, which subsequently leads to the recruitment and activation of hepatic stellate cells (HSCs) as well as the excess production of extracellular matrix (ECM) proteins [2, 5, 6]. In addition, inflammation acts as a fuel to accelerate liver cell proliferation and tissue regeneration. During fibrotic liver diseases, diverse hepatic immune cells, especially macrophages, are dynamically recruited to the injury site in a manner mainly determined by the cytokines and chemokines (C-C motif chemokine ligand 2 [CCL2], C-C motif chemokine ligand 5 [CCL5], C-X-C motif chemokine ligand 10 [CXCL10], etc.) secreted by hepatocytes, HSCs, and endothelial cells [2]. These inflammatory factors control the migration and positioning of immune cells and HSCs, which express chemokine receptors, thus defining the magnitude of the inflammatory response during fibrosis progression. Among these chemokines, CCL2 is the most widely studied. The important role of the CCL2-C-C motif chemokine receptor 2 [CCR2] signaling in liver fibrosis has been established in several experimental models using CCL2- or CCR2-deficient mice. The functional relevance of CCL2 is dependent on the recruitment of HSCs and infiltration of macrophages [7–10]. Another critical chemokine pathway is the CCL5 (also known as regulated upon activation normal T cell expressed and secreted factor [RANTES])-CCR1/CCR5 pathway, which is largely enhanced in fibrotic livers [11]. Either genetic knockout of CCL5 or pharmacological inhibition of CCL5 by the antagonist Met-CCL5 (CCL5 protein Ser24-Ser91, with an N-terminal Met) in mice ameliorated experimental liver fibrosis [12]. In addition, CXCL10 seems profibrogenic, either by modulating hepatic macrophage infiltration or by inhibiting natural killer (NK) cell–mediated HSC inactivation [13, 14]. Although the downstream effects of these chemokines are well defined, the associated upstream signaling and transcriptional regulation in hepatocytes remain largely unknown.

Mediator is a transcriptional cofactor complex that is considered as part of the general transcriptional machinery. In response to environmental or developmental cues, Mediator can activate or repress specific gene transcription through physically interacting with distinctive DNA-bound transcription factors (TFs) [15, 16]. Increasing evidence suggests that different Mediator subunits control diverse signaling pathways and biological processes; and its dysregulation leads to developmental abnormalities, metabolic disorders, and cancer [17]. In this study, we focused on Mediator complex subunit 23 (MED23), which has been shown to participate in hepatic glucose and lipid metabolism [18]. Although mice with liver-specific knockout of *Med23* (*med23*$^{\Delta li}$) do not display abnormal hepatic histology and function, *Med23* ablation improves hepatic glucose metabolism, as reflected by enhanced glucose tolerance and insulin sensitivity. Notably, liver-specific *Med23* deletion prevented high-fat diet (HFD)- and genetic-induced obesity and reduced the related pathological consequences of this condition [18]. The spectrum of liver diseases could range from simple steatosis to nonalcoholic steatohepatitis (NASH), fibrosis, cirrhosis, and liver cancer [19]. Considering the connection between liver metabolism and chronic liver diseases, we sought to determine whether MED23 might be involved in the pathogenesis of liver fibrosis, and we found that the silencing of *Med23* in hepatocytes aggravates the development of carbon tetrachloride (CCl$_4$)-induced hepatic inflammation and fibrosis, thus defining an important role of hepatic MED23 in liver fibrosis. We further unraveled a novel molecular mechanism by which hepatic MED23 negatively regulates the expression of the hepatic chemokines CCL5 and CXCL10 via suppressing the activity of RAR-related orphan receptor alpha (RORα). RORα has been shown in previous studies to be involved in cerebellum development, circadian rhythms, lipid metabolism, and fat accumulation [20–24]. Our study demonstrated that hepatic RORα, in cooperation with MED23, plays a role in inflammatory responses, acting as a positive regulator of CCL5 and CXCL10 in

metalloproteinase; Tnfα, tumor necrosis factor alpha; WT, wild-type.

initiating the liver fibrosis, which suggests new molecular targets for clinical intervention in liver fibrosis.

## Results

### Mice with hepatocyte-specific *Med23* ablation exhibit enhanced liver fibrosis

To investigate the role of MED23 in liver fibrosis, we employed the well-established CCl$_4$-induced mouse model of liver fibrosis [25], in which the control *med23*-floxed (*med23*$^{f/f}$) mice and mice with hepatocyte-specific *Med23* deletion (*med23*$^{\Delta li}$) [18] were intraperitoneally injected with CCl$_4$ every 3 days for 1 month. After chronic administration of CCl$_4$, *med23*$^{\Delta li}$ mouse livers exhibited augmented collagen deposition and activated HSC (myofibroblast) expansion compared with these characteristics in *med23*$^{f/f}$ mouse livers, reflected by the increases in Sirius red–positive areas and alpha-smooth muscle actin (α-SMA)-positive areas (Fig 1A and 1B). Increased α-SMA expression in *med23*$^{\Delta li}$ mice was further confirmed by immunoblotting and quantitative real-time PCR (qRT-PCR) of whole-liver homogenates (*Acta2* encodes α-SMA) (Fig 1C and 1D).

Additionally, the expression of *Desmin*, a marker of HSCs reflecting the number of HSCs, was also higher in *med23*$^{\Delta li}$ mouse livers than in *med23*$^{f/f}$ mouse livers (Fig 1D). Consistent with the augmented collagen deposition, the collagen type I alpha 1 chain (*Col1a1*) and collagen type III alpha 1 chain (*Col3a1*) levels were up-regulated after *Med23* ablation in hepatocytes (Fig 1D). As liver fibrosis is accompanied by dynamic ECM remodeling [1], we further found that the transcription of matrix metalloproteinases (MMPs) and MMP inhibitors (tissue inhibitors of metalloproteinases [TIMPs]), including *Mmp9*, *Mmp13*, *Timp1*, and *Timp2*, were higher in *med23*$^{\Delta li}$ mouse livers than in control mouse livers (Fig 1E). In agreement with the increased exacerbation of liver fibrosis, the mRNA levels of profibrogenic factors such as transforming growth factor beta 1 (*Tgfβ1*) and platelet derived growth factor beta (*Pdgfβ*) and their receptors were significantly higher in *med23*$^{\Delta li}$ mouse livers than in *med23*$^{f/f}$ mouse livers (Fig 1F). Collectively, these data suggest that the ablation of *Med23* in hepatocytes renders mice prone to liver fibrosis.

We also examined the effect of hepatic *Med23* deficiency in another NASH model, which used a methionine and choline-deficient (MCD) diet to induce liver pathology including steatosis, liver inflammation and fibrosis, and hepatocyte death (Wang and colleagues, 2016). Similar to the CCl$_4$-induced fibrosis model, *Med23* silencing also resulted in increasing mRNAs related to liver fibrosis, including *Col1a1*, *Col3a1*, *Tgfβ1*, *Pdgfβ*, *Tgfβr1*, *Pdgfrβ*, and *Timp1* (S1A Fig). Importantly we observed that there was an increase in α-SMA$^+$ cells (S1B and S1C Fig) as well as protein level of Col1a1 in *Med23*-deficient mouse livers (S1D and S1E Fig), suggesting that *Med23* ablation can also promote MCD diet–induced liver fibrosis.

### *Med23*$^{\Delta li}$ mice display enhanced hepatocyte proliferation after the chronic administration of CCl$_4$

Liver fibrosis is generally considered the consequence of wound responses accompanied by hepatocyte proliferation shortly after injury-induced cell death [3]. Therefore, we next examined cell death and proliferation in *med23*$^{f/f}$ and *med23*$^{\Delta li}$ mouse livers after CCl$_4$ exposure. TUNEL staining revealed fewer apoptotic cells in *med23*$^{\Delta li}$ mouse livers than in control mouse livers (Fig 2A and 2B). Consistent with this result, the expression of the antiapoptotic genes cellular inhibitor of apoptosis protein 1 (*cIAP1*) and cellular inhibitor of apoptosis protein 2 (*cIAP2*) was slightly increased in livers with *Med23* deletion compared with that in control

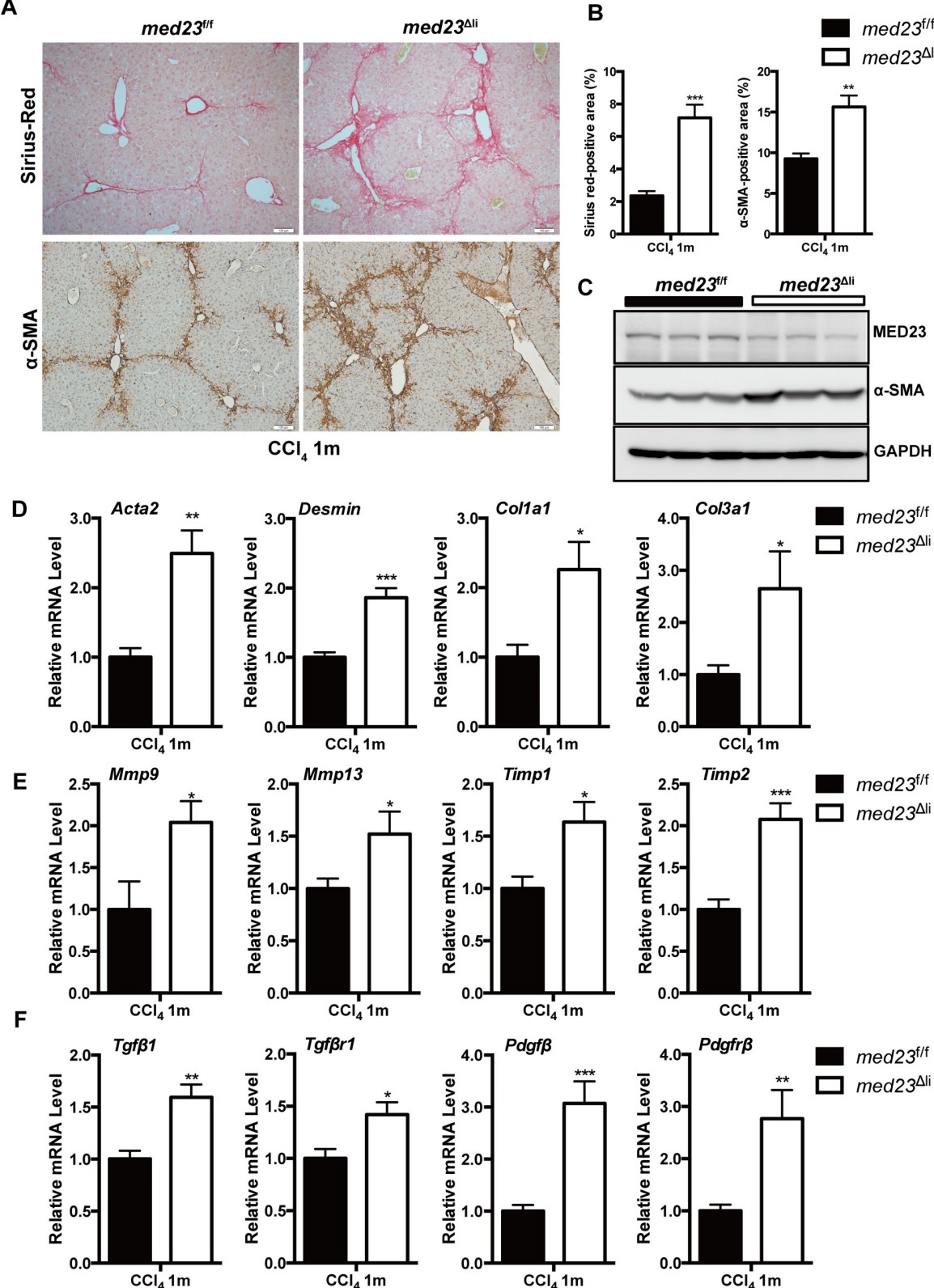

**Fig 1. Analysis of liver fibrosis in *med23*^f/f^ and *med23*^Δli^ mice after chronic administration of CCl₄.** (A) Histology analysis of livers from *med23*^f/f^ and *med23*^Δli^ mice. The liver sections of *med23*^f/f^ and *med23*^Δli^ mice after chronic CCl₄ administration were stained with Sirius red and α-SMA. Representative pictures were shown. (B) Quantification of Sirius red–positive ($n = 5$ per group) and α-SMA-

positive areas ($n$ = 7 per group) in the liver sections of $med23^{f/f}$ and $med23^{\Delta li}$ mice. (C) The total protein was extracted from whole livers of $med23^{f/f}$ and $med23^{\Delta li}$ mice and analyzed by western blotting using the indicated antibodies. GAPDH was used as a loading control. (D-F) The total RNA was extracted from whole livers of $med23^{f/f}$ and $med23^{\Delta li}$ mice and then analyzed by qRT-PCR to detect the expression of the liver fibrosis-associated genes. The expression was normalized to $Gapdh$ ($n$ = 7 per group). Data are presented as means ± SEM. $^{*}P < 0.05$, $^{**}P < 0.01$, $^{***}P < 0.001$. For underlying data, see S1 Data file. α-SMA, alpha-smooth muscle actin; CCl$_4$, carbon tetrachloride; Col, collagen; GAPDH, glyceraldehyde 3-phosphate dehydrogenase; $med23$, Mediator complex subunit 23; $med23^{\Delta li}$, liver-specific knockout of $Med23$; $med23^{f/f}$, $med23$-floxed; Mmp, matrix metalloproteinase; Pdgfβ, platelet derived growth factor beta; Pdgfrβ, platelet-derived growth factor receptor beta; qRT-PCR, quantitative real-time PCR; Tgfβ1, transforming growth factor beta 1; Tgfβr1, transforming growth factor beta receptor 1; Timp, tissue inhibitor of metalloproteinase.

livers (Fig 2C). Interestingly, we noticed that the apoptotic cells were generally localized with the α-SMA-positive areas in both experimental groups (Fig 2A), suggesting that the observed dead cells were mainly HSCs and that the increased myofibroblast numbers in $med23^{\Delta li}$ mouse livers could be the consequence of decreased cell death. In addition, serum alanine amino-transferase (ALT) and aspartate aminotransferase (AST) secretion by damaged hepatocytes, as well as albumin production, were equal in the experimental groups (S2A Fig), suggesting that the degree of liver injury was similar. The liver weight was slightly decreased in $med23^{\Delta li}$ mice, whereas the liver/body weights of $med23^{\Delta li}$ mice were comparable to those of $med23^{f/f}$ mice after chronic CCl$_4$ administration (S2B Fig). Noticeably, there was a dramatic increase in hepa-tocyte proliferation after $Med23$ ablation, as evidenced by the increased number of proliferat-ing cell nuclear antigen (PCNA)-positive hepatocytes compared with that in control mice (Fig 2D). Consistently, qRT-PCR analysis of mRNA from whole-liver homogenates demonstrated that the level of hepatocyte growth factor ($Hgf$), a key extracellular factor for hepatocyte growth, was higher in $med23^{\Delta li}$ livers than in control livers (Fig 2E). Moreover, compared with $med23^{f/f}$ mice, $med23^{\Delta li}$ mice exhibited a significant increase in the expression of genes associ-ated with cell proliferation, including $Cyclin\ D1$, $c$-$Fos$, and $c$-$Jun$ (Fig 2E). Taken together, these results indicate that $med23^{\Delta li}$ mice have improved resistance to CCl$_4$-induced hepatic cell death, which may explain the enhanced fibrosis. In addition, $med23^{\Delta li}$ mouse livers dis-played greater hepatocyte proliferation than control mouse livers, implying an improvement in the repair capability.

## Inflammatory infiltration is increased in $med23^{\Delta li}$ mice compared with that in controls after chronic administration of CCl$_4$

The infiltration of numerous immune cell populations, including monocytes/macrophages and T cells [2], into the liver is a crucial pathogenic feature following acute and chronic liver injury. Therefore, we next sought to analyze the differential immune responses in the liver between the experimental groups. Histological analysis showed a clear increase in immune cell infiltration (Fig 3A), which was further confirmed by the increased expression of the pan-leu-kocyte markers $Cd45$ and $Cd3g$ (Fig 3C) in $med23^{\Delta li}$ mice compared with that in control mice after chronic CCl$_4$ treatment. Specifically, we found that the population of macrophages in the liver, as indicated by immunostaining and qRT-PCR analysis of F4/80 expression, was strik-ingly increased in $Med23$-deficient livers compared with that in control livers (Fig 3A–3C). We further analyzed hepatic mRNA expression of proinflammatory cytokines and chemokines that are responsible for guiding the migration of various immune cells. Whereas the hepatic tumor necrosis factor alpha ($Tnf\alpha$) and interleukin 6 ($Il$-$6$) mRNA levels were comparable between the experimental groups, the levels of $Il$-$1\alpha$ and $Il$-$1\beta$ were dramatically increased in $med23^{\Delta li}$ mice compared with those in $med23^{f/f}$ mice (Fig 3D). Furthermore, we observed increases of at least 1.5-fold in the levels of chemokines, including $Ccl2$, $Ccl4$, $Ccl5$, and $Cxcl10$, as well as the levels of chemokine receptors, including $Ccr1$, $Ccr2$, $Ccr5$, and C-X-C motif

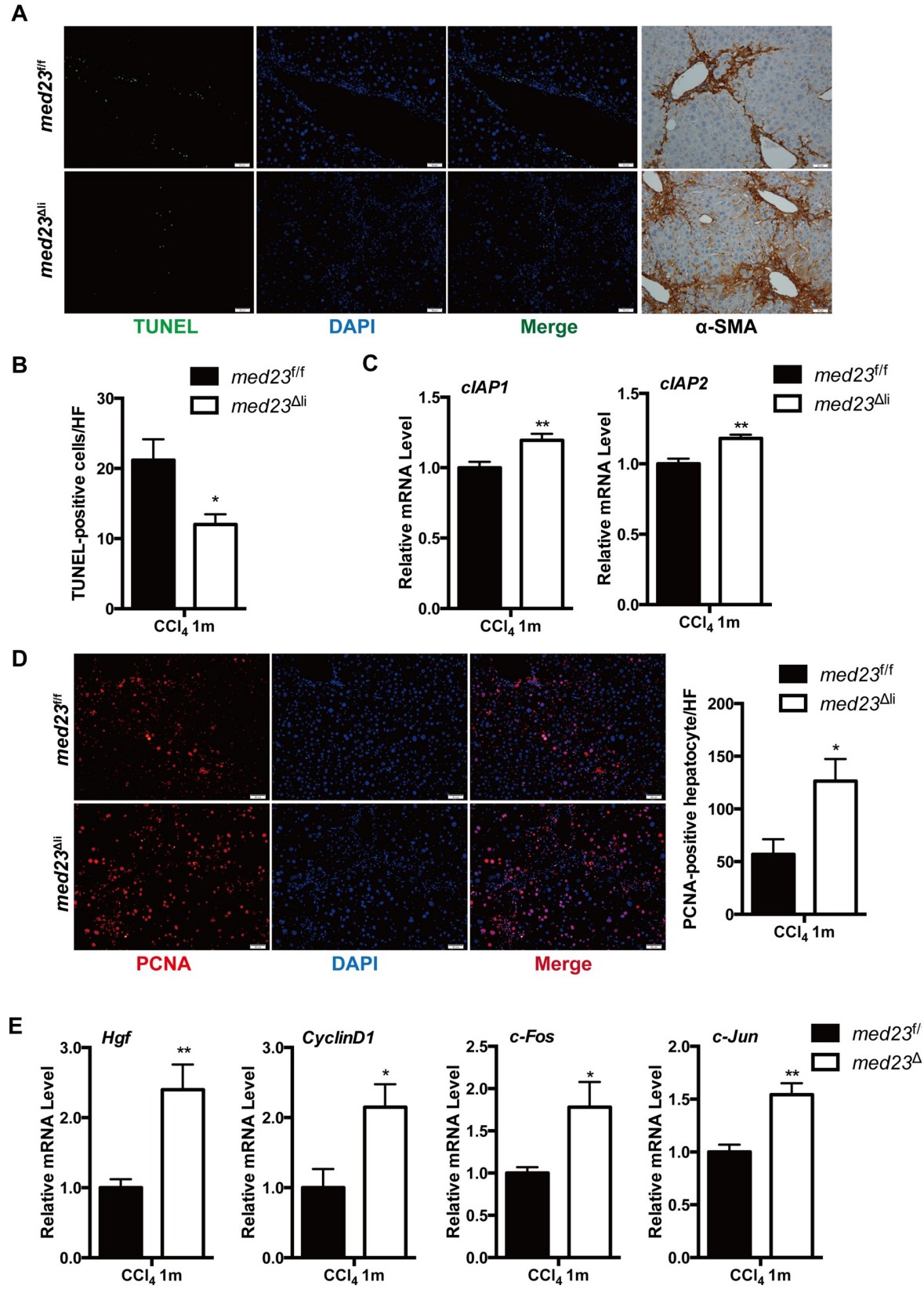

**Fig 2. Reduced liver cell death and enhanced hepatocyte proliferation in *med23*<sup>Δli</sup> mice after chronic administration of CCl₄.** (A) Representative views of TUNEL staining and α-SMA staining in the liver sections of *med23*^f/f^ and *med23*^Δli^ mice. (B) Quantification of TUNEL-positive cells from the liver sections of *med23*^f/f^ and *med23*^Δli^ mice (*med23*^f/f^, $n = 6$; *med23*^Δli^, $n = 7$). (C) The expression of antiapoptotic genes (*cIAP1* and *cIAP2*) in whole-liver extracts was analyzed by qRT-PCR ($n = 7$ per group). (D) Representative views and quantification of PCNA staining in the liver sections of *med23*^f/f^ and *med23*^Δli^ mice ($n = 7$ per group). (E) qRT-PCR analysis of proliferative genes (*Hgf*, *CyclinD1*, *c-Fos*, and *c-Jun*) in whole-liver extracts of *med23*^f/f^ and *med23*^Δli^ mice ($n = 7$ per group). Data are presented as means ± SEM. *$P < 0.05$, **$P < 0.01$. For underlying data, see S1 Data file. α-SMA, alpha-smooth muscle actin; CCl₄, carbon tetrachloride; cIAP, cellular inhibitor of apoptosis protein; Hgf, hepatocyte growth factor; med23, Mediator complex subunit 23; *med23*^Δli^, liver-specific knockout of *Med23*; *med23*^f/f^, med23-floxed; PCNA, proliferating cell nuclear antigen; qRT-PCR, quantitative real-time PCR.

chemokine receptor 2 (*Cxcr2*), in *med23*^Δli^ mice relative to these levels in *med23*^f/f^ mice (Fig 3E and 3F). Thus, mice with *Med23* deletion in hepatocytes seem to be able to mount augmented immune responses to chronic CCl₄ administration, which possibly results in enhanced liver fibrosis and increased compensatory hepatocyte proliferation.

## *Med23* deletion enhances the secretion of proinflammatory cytokines and chemokines after acute CCl₄ treatment

To further address the causal associations among the Mediator subunit MED23 expression, proinflammatory factors production, and macrophage infiltration, we characterized the acute (24-hour) response to CCl₄ administration in *med23*^f/f^ and *med23*^Δli^ mouse livers. It is known that CCl₄-induced hepatotoxicity is characterized by centrilobular necrosis [26]. Substantial necrosis was observed in control livers by hematoxylin–eosin (HE) staining (Figs 4A and S3A), but necrosis was scarcely detected in *med23*^Δli^ livers at 24 hours after injury (Figs 4A and S3A). Consistent with the results of chronic CCl₄ treatment, TUNEL staining showed much fewer apoptotic cells in acutely injured *med23*^Δli^ mouse livers than in *med23*^f/f^ mouse livers (Fig 4A and 4B); this result was verified by the decreased levels of serum ALT and AST (S3B Fig) as well as cleaved caspase 3 and phosphorylation of the histone variant H2AX (γ-H2AX) on immunoblots of whole-liver protein extracts (Fig 4C), suggesting that *Med23* ablation may render the liver refractory to the CCl₄-induced hepatic damage. Although acute CCl₄ treatment for 24 hours was not able to induce hepatocyte proliferation, as evidenced by the very small numbers of Ki67-positive hepatocytes in control mice, we did observe a small number of proliferative hepatocytes in *med23*^Δli^ mice (Fig 4D and 4E). Moreover, qRT-PCR analysis of whole-liver homogenates indicated elevated expression of the proliferation-promoting gene *c-Myc* in *med23*^Δli^ mice compared with that in control mice (Fig 4F). These data suggest that liver-specific ablation of the Mediator subunit *Med23* gene resulted in reduced liver damage and enhanced compensatory hepatocyte proliferation in response to acute CCl₄ treatment.

We next performed α-SMA and F4/80 immunostaining as well as a qRT-PCR assay to determine the number of activated HSCs and macrophages, respectively. In contrast to our findings for chronic CCl₄ treatment, we failed to detect a difference in the numbers of activated HSCs and macrophages between acutely injured control mice and *med23*^Δli^ mice (S3C–S3F Fig). However, we did observe higher mRNA transcript levels of proinflammatory cytokines, including *Tnfα*, *Il-6*, and *Il-1β*, as well as chemokines, including *Ccl2*, *Ccl3*, *Ccl4*, *Ccl5*, *Ccl7*, and *Cxcl10*, in acutely injured *med23*^Δli^ mice than in control mice (Fig 4G and 4H), suggesting that these increased cytokines/chemokines in *med23*^Δli^ livers may be not produced by macrophages and HSCs, because the numbers of both of these cell types did not differ between control and *med23*^Δli^ livers after 24 hours of CCl₄ treatment. Specifically, the protein level of CCL2, which is important for monocyte chemotaxis, was also higher in *med23*^Δli^ mice than in control mice (S3G Fig). By contrast, there was no significant change in the expression of the profibrogenic factors *Tgfβ1* and *Pdgfβ* or the fibrogenic factors *Col1a1* and *Col3a1* between the

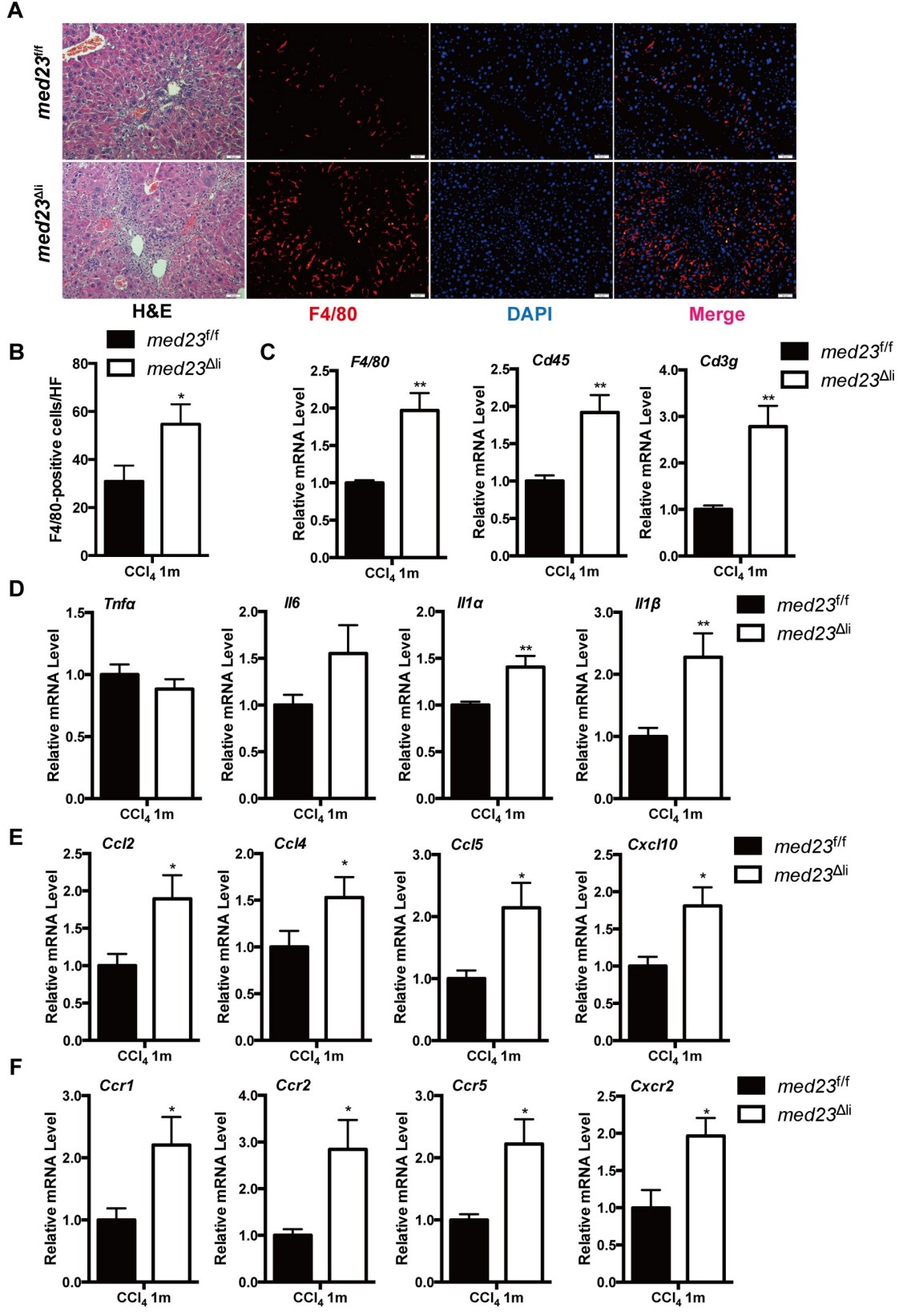

**Fig 3. Increased inflammatory infiltration in *med23*^Δli mice after the chronic administration of CCl₄.** (A) Liver sections from *med23*^f/f and *med23*^Δli mice were stained with HE, F4/80, and DAPI respectively. (B) Quantification of F4/80-positive cells from the liver sections of *med23*^f/f and *med23*^Δli mice (*med23*^f/f, $n = 5$; *med23*^Δli, $n = 7$). (C) The mRNA expression of immune cells markers (*F4/80*, *Cd45*, and *Cd3g*) in whole-liver extracts was analyzed by qRT-PCR ($n = 7$ per group). (D-F) qRT-PCR analysis of the expression of cytokines, chemokines, and their receptors in the liver ($n = 7$ per group). Data are presented as means ± SEM. $^*P < 0.05$, $^{**}P < 0.01$. For underlying data, see S1 Data file. Ccl, C-C motif chemokine ligand; CCl₄, carbon tetrachloride; Ccr, C-C motif chemokine receptor; Cxcl, C-X-C motif chemokine ligand; Cxcr, C-X-C motif chemokine receptor; HE, hematoxylin–eosin; Il, interleukin; *med23*, Mediator complex subunit 23; *med23*^Δli, liver-specific knockout of *Med23*; *med23*^f/f, *med23*-floxed; qRT-PCR, quantitative real-time PCR; *Tnfα*, tumor necrosis factor alpha.

experimental groups at this early time point (S3H Fig). Thus, it seems that hepatocyte-specific deletion of *Med23* leads to increased secretion of proinflammatory cytokines and chemokines initially, which may subsequently facilitate macrophage and leukocyte recruitment as well as HSC activation.

## Hepatic MED23 suppresses *Ccl5* and *Cxcl10* expression in vivo and in vitro

Hepatic macrophages are well characterized as a main source of proinflammatory factors that contribute to liver fibrosis [2]. However, hepatocytes have also been suggested to produce proinflammatory factors during the process of liver fibrosis [27]. Because it is not clear whether the increased inflammatory factors are derived from the *Med23*-deficient hepatocytes or the infiltrated immune cells, we depleted liver macrophages by tail vein injection of clodronate liposomes (Fig 5A), as reported [28, 29]. Indeed, compared with injection of control liposomes, injection of clodronate liposomes into *med23*^Δli mice resulted in an evident reduction in macrophage numbers, as indicated by the expression of the surface marker F4/80, and it did not affect the efficiency of *Med23* deletion (Fig 5B and 5C). Clodronate administration also abolished the marked increase in the production of some cytokines and chemokines, including *Tnfα*, *Il-6*, *Il-1α*, *Il-1β*, and *Ccl2*, in *Med23*-deficient livers under acute CCl₄ treatment (Fig 5D), suggesting that macrophages may be the source of these increased inflammatory factors. However, *Ccl5* and *Cxcl10* expression remained higher in *med23*^Δli mice than in control mice after injection with clodronate liposomes (Fig 5D), suggesting that the increased levels of these two important chemokines could be produced by hepatocytes instead of macrophages.

These phenomena seemed to be cell autonomous because *Med23* knockdown dramatically enhanced the transcriptional levels of some proinflammatory factors, especially *Ccl5* and *Cxcl10*, in alpha mouse liver 12 (AML12) immortalized mouse hepatocytes (Fig 5E). These regulatory effects were confirmed in *Med23*-knockout AML12 cells created by clustered regularly interspaced short palindromic repeats/CRISPR-associated protein 9 (CRISPR/Cas9) (Fig 5F and 5G). We further tested the inhibitory role of the Mediator subunit MED23 in transient transfection assays; surprisingly, increasing amounts of *Med23* overexpression increased the mRNA level of *Ccl5* and *Cxcl10* in a dose-dependent manner (Fig 5H), but did not affect the expression levels of *Tnfα* or *Ccl2*, which are independent of hepatic MED23 expression (Fig 5H). By contrast, overexpression of another Mediator subunit, MED24, did not greatly change the expression of *Ccl5* or *Cxcl10* (Fig 5I), underscoring the specific effect of MED23 on *Ccl5* and *Cxcl10* expression. A previous study proposed that *Med23* overexpression could inhibit transcriptional activation through sequestering the whole Mediator complex from the transcriptional machinery [30]. Here, it seems that the overexpressed MED23 is able to sequester the whole Mediator complex, thus counteracting the inhibitory effect of the Mediator complex on *Ccl5* and *Cxcl10* expression.

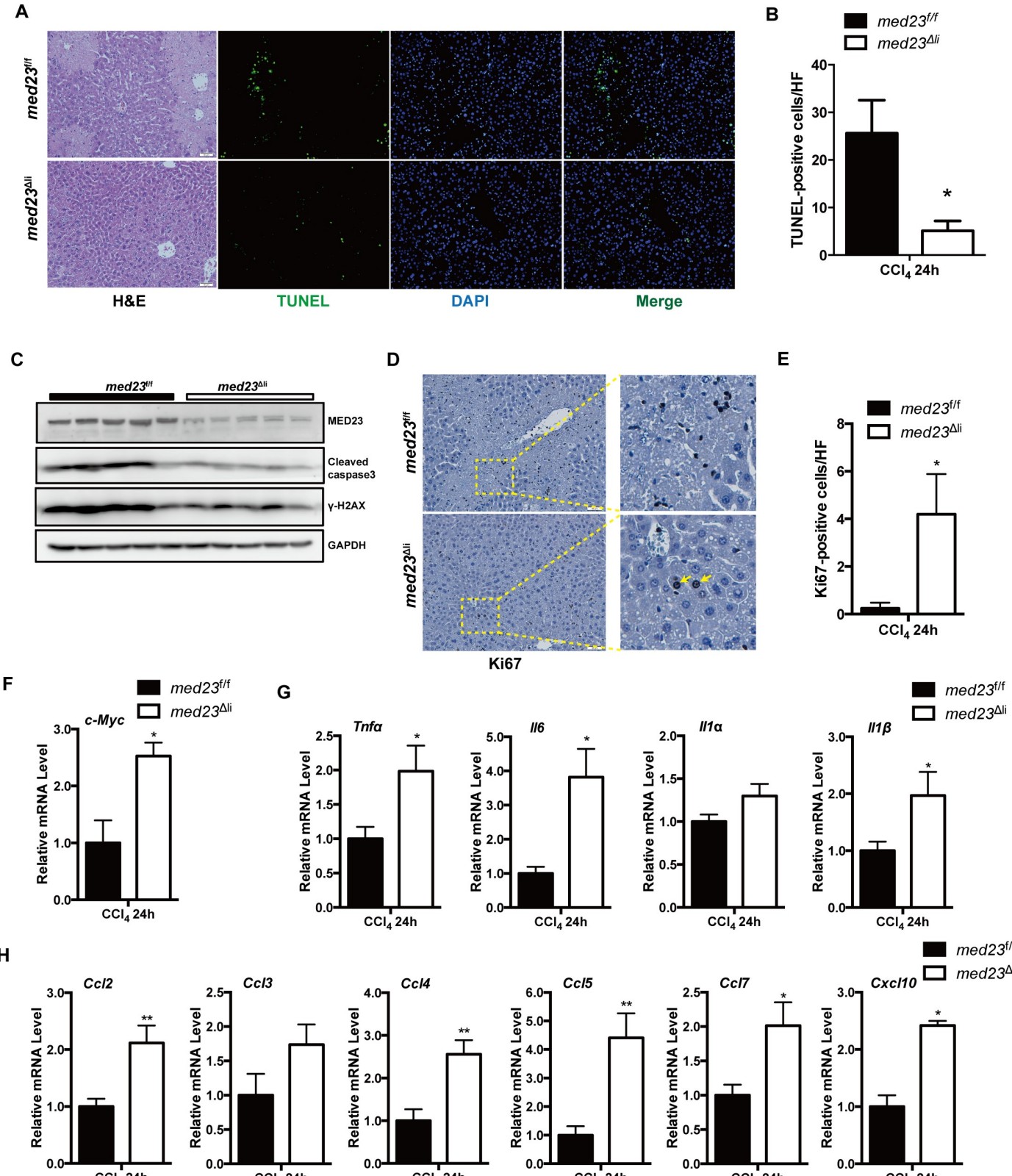

**Fig 4. More proinflammatory cytokine and chemokine secretion after acute administration of CCl₄.** (A) Representative views of HE, TUNEL, and DAPI staining in the liver sections of *med23*^f/f^ and *med23*^Δli^ mice after acute CCl₄ administration for 24 hours. (B) Quantification of TUNEL-positive cells from the liver

sections of *med23*<sup>f/f</sup> and *med23*<sup>Δli</sup> mice (*med23*<sup>f/f</sup>, *n* = 6; *med23*<sup>Δli</sup>, *n* = 5). (C) The total protein was extracted from whole livers of *med23*<sup>f/f</sup> and *med23*<sup>Δli</sup> mice and analyzed by western blotting using the indicated antibodies. GAPDH was used as a loading control. (D, E) Representative views of Ki67 staining (D) and quantification (E) in the liver sections of *med23*<sup>f/f</sup> and *med23*<sup>Δli</sup> mice (*med23*<sup>f/f</sup>, *n* = 5; *med23*<sup>Δli</sup>, *n* = 4). (F) qRT-PCR analysis of *c-Myc* in whole-liver extracts of *med23*<sup>f/f</sup> and *med23*<sup>Δli</sup> mice (*med23*<sup>f/f</sup>, *n* = 6; *med23*<sup>Δli</sup>, *n* = 5). (G, H) qRT-PCR analysis of cytokines and chemokines in whole-liver extracts of *med23*<sup>f/f</sup> and *med23*<sup>Δli</sup> mice (*med23*<sup>f/f</sup>, *n* = 6; *med23*<sup>Δli</sup>, *n* = 5). Data are presented as means ± SEM. *$P$ < 0.05, **$P$ < 0.01. For underlying data, see S1 Data file. γ-H2AX, phosphorylation of the histone variant H2AX; Ccl, C-C motif chemokine ligand; CCl₄, carbon tetrachloride; Cxcl, C-X-C motif chemokine ligand; GAPDH, glyceraldehyde 3-phosphate dehydrogenase; HE, hematoxylin–eosin; Il, interleukin; *med23*, Mediator complex subunit 23; *med23*<sup>Δli</sup>, liver-specific knockout of *Med23*; *med23*<sup>f/f</sup>, *med23*-floxed; qRT-PCR, quantitative real-time PCR; *Tnfα*, tumor necrosis factor alpha.

## Hepatocyte MED23 cooperates with RORα to regulate *Ccl5* and *Cxcl10* expression

To better understand the mechanisms through which *Med23* deficiency enhances *Ccl5* and *Cxcl10* expression in hepatocytes, we carried out a genome-wide transcriptome analysis of whole livers from control and *med23*<sup>Δli</sup> mice 24 hours after CCl₄ treatment. RNA-seq data analysis revealed that 593 genes were down-regulated more than 2-fold and 660 genes were up-regulated more than 2-fold in the *med23*<sup>Δli</sup> set relative to their expression in the control set (Fig 6A). Gene set enrichment analysis (GSEA) of these up-regulated genes showed that they were enriched in monocyte chemotaxis, consistent with the effect of *Med23* deletion in up-regulating the expression of various chemokines (Fig 6B and 6C).

Mediator is known as a transcriptional cofactor targeted by distinct TFs in response to different signaling stimulators [15]. To identify the potential TFs cooperating with MED23 to control the expression of various chemokines, we then performed Ingenuity Pathway Analysis (IPA) on significantly up-regulated genes with a 2-fold increase in expression after *Med23* deletion in hepatocytes. The predicted upstream regulators pointed toward several TFs, including RORα, CCAAT enhancer binding protein beta (C/EBPβ), RORγ, and peroxisome proliferator activated receptor alpha (PPARα), possibly activated after *Med23* ablation in hepatocytes (Fig 6D), implying that these genes could contribute to the increased chemokine expression. Analysis of the published database revealed higher *RORα* expression in cirrhosis samples than in normal samples, whereas the expression of other TFs was unchanged (S4A Fig); therefore, we focused on further analysis of RORα. To verify that RORα regulates *Ccl5* and *Cxcl10* expression, we knocked down *Rorα* in wild-type (WT) AML12 cells using small interfering RNA (siRNA) and observed that the *Ccl5* and *Cxcl10* mRNA levels were consistently decreased after RNA interference (RNAi) transfection (Fig 6E). The protein level of CXCL10 was much lower in *Rorα*-silenced AML12 cells than in control cells (Fig 6F). We further introduced *RORα* into the WT AML12 cells using retroviral transduction and found that ectopic *RORα* expression indeed activated RORα response element (RORE) reporter activity (Figs 6G and S4B). qRT-PCR and luciferase reporter results revealed that ectopic expression of *RORα* strongly stimulated *Ccl5* expression and modestly increased *Cxcl10* expression (Fig 6G and 6H) but had no effect on *Ccl2* transcription level (S4C Fig). These data suggested that RORα is a novel upstream regulator of *Ccl5* and *Cxcl10*. To further understand how MED23 cooperates with RORα to regulate *Ccl5* and *Cxcl10* expression, *Rorα* was knocked down in AML12 cells with or without *Med23* knockdown. Notably, the increase in the levels of *Ccl5* and *Cxcl10* mRNA by *Med23* deficiency was significantly reduced after *Rorα* knockdown (Fig 6I), suggesting a possible functional interaction between RORα and MED23 in controlling *Ccl5* and *Cxcl10* expression but not *Ccl2* expression (S4D Fig).

We next performed dual-luciferase reporter assays to assess the effect of MED23 on RORα-driven RORE reporter activity. Consistent with endogenous *Ccl5* and *Cxcl10* regulation, *MED23* deletion significantly increased the RORE reporter activity driven by RORα (Fig 6J). Moreover, we transfected HeLa cells with increasing dosage of *RORα* and *Med23* along with

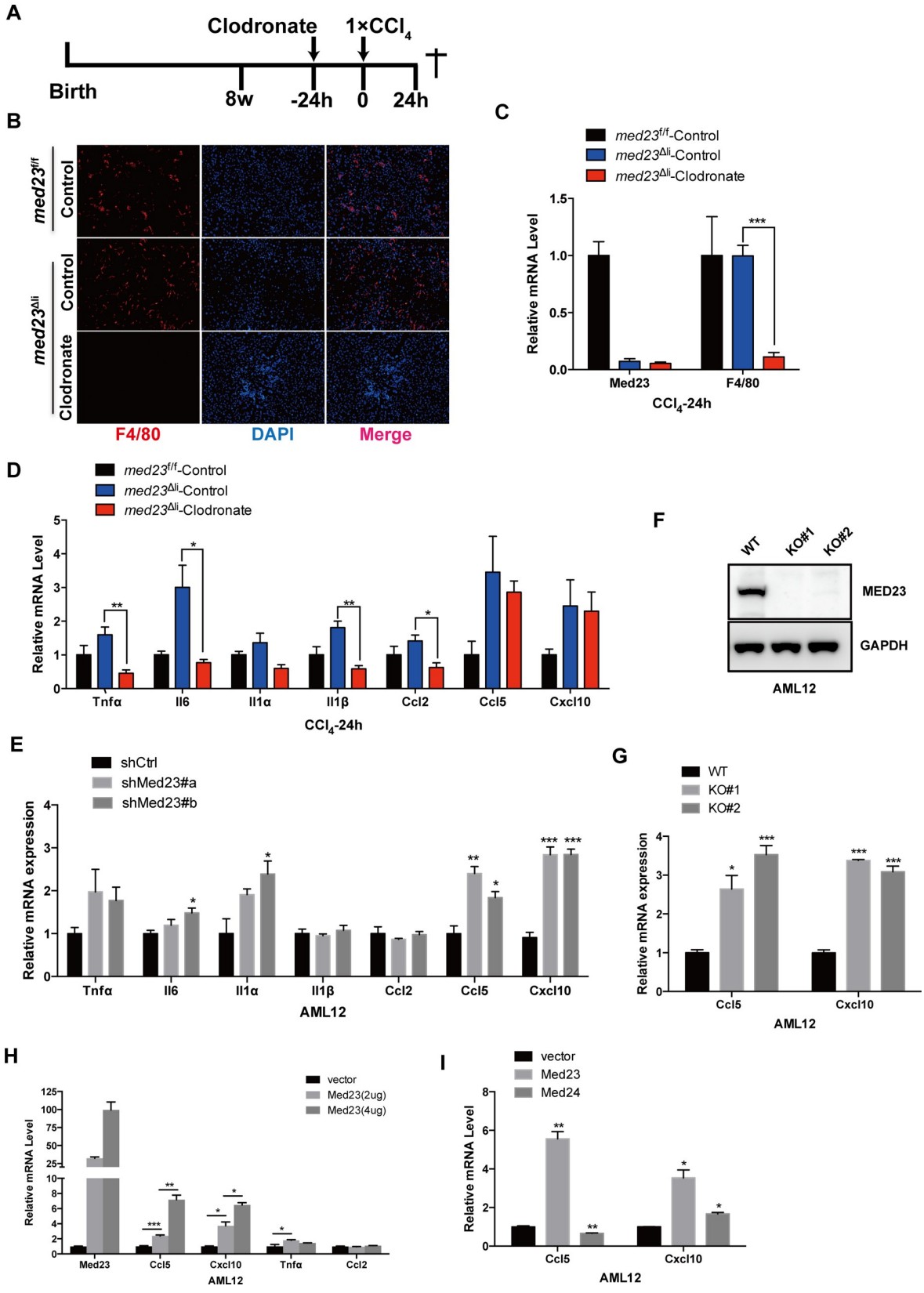

**Fig 5. Increased *Ccl5* and *Cxcl10* expression directly in hepatocytes with *Med23* deletion in vivo and in vitro.** (A) Strategy to delete macrophages in liver and administrate with acute CCl$_4$ for 24 hours. (B) Representative views of F4/80 staining in the liver sections of *med23*$^{f/f}$ and *med23*$^{\Delta li}$ mice after tail intravenous injection of control liposome and clodronate liposome and then CCl$_4$ administration for 24 hours. (C) qRT-PCR analysis of *Med23* and *F4/80* in whole-liver extracts of *med23*$^{f/f}$ and *med23*$^{\Delta li}$ mice (*med23*$^{f/f}$-Control, $n = 4$; *med23*$^{\Delta li}$-Control, $n = 4$; *med23*$^{\Delta li}$-Clodronate, $n = 7$). (D, E) qRT-PCR analysis of proinflammatory cytokines and chemokines in whole-liver extracts of *med23*$^{f/f}$ and *med23*$^{\Delta li}$ mice (*med23*$^{f/f}$-Control, $n = 4$; *med23*$^{\Delta li}$-Control, $n = 4$; *med23*$^{\Delta li}$-Clodronate, $n = 7$) (D) and AML12 cells with *Med23* knockdown ($n = 3$ per group) (E). (F) The protein level of MED23 was analyzed by western blotting in AML12 cells after CRISPR/Cas9 mediated *Med23* KO. GAPDH was used as a loading control. (G) qRT-PCR analysis of *Ccl5* and *Cxcl10* in *Med23* KO AML12 cells ($n = 3$ per group). (H, I) qRT-PCR analysis of proinflammatory cytokines and chemokines in AML12 cells after transient transfection ($n = 3$ per group). Data are presented as means ± SEM. $^*P < 0.05$, $^{**}P < 0.01$, $^{***}P < 0.001$. For underlying data, see S1 Data file. AML12, alpha mouse liver 12; Ccl, C-C motif chemokine ligand; CCl$_4$, carbon tetrachloride; Cxcl, C-X-C motif chemokine ligand; GAPDH, glyceraldehyde 3-phosphate dehydrogenase; Il, interleukin; KO, knockout; MED23, Mediator complex subunit 23; *med23*$^{\Delta li}$, liver-specific knockout of *Med23*; *med23*$^{f/f}$, *med23*-floxed; qRT-PCR, quantitative real-time PCR; shCtrl, negative control vector containing scrambled shRNA; shMed23, shRNA against Med23; *Tnfα*, tumor necrosis factor alpha; WT, wild-type.

RORE reporter to further verify the regulation of RORα activity by MED23. Increasing RORα expression up-regulates the RORE reporter activity in a dose-dependent manner (Fig 6K). Importantly, RORα-activated RORE activity was repressed gradually with increasing dosage of *Med23* (up to 2-fold decrease) (Fig 6K). Together, these data suggested that the Mediator subunit MED23 could act as a "corepressor" of RORα in directing *Ccl5* and *Cxcl10* transcription.

Mediator is well recognized as a coactivator for transcriptional activation; however, there are a few studies showing that several Mediator subunits, especially the kinase submodule, may also inhibit gene transcription [31–33]. For example, Mediator has been shown to link the RE1 silencing TF (REST) with the G9a histone methyltransferase to suppress neuronal gene expression through histone H3K9 dimethylation in nonneuronal cells [31, 34]. To dissect how the Mediator subunit MED23 suppresses cytokine gene expression, we performed chromatin immunoprecipitation (ChIP) assays and found that the binding pattern of MED23 and RNA polymerase II (Pol II) at the promoter regions (P1, P2, and P3) of *Ccl5* and *Cxcl10* were similar; importantly, the occupancy of MED23 is most enriched at the core promoter regions (P1 and P2) (Fig 7A). Interestingly we also observed the occupancy of MED23 in the *Ccl5* coding region (P4), implying that MED23 might also affect *Ccl5* expression via the postrecruitment function (Fig 7A). We further analyzed the histone methylation profiles of the *Ccl5* and *Cxcl10* gene promoters in *med23*$^{f/f}$ and *med23*$^{\Delta li}$ mouse liver tissues and observed high H3K9 dimethylation occupancy in the proximal and distal promoters of *Ccl5* and *Cxcl10*, which was largely reduced in livers with *Med23* deletion (Fig 7B). In comparison, the H3K4 trimethylation occupancy in the *Ccl5* and *Cxcl10* promoters was comparable between *med23*$^{f/f}$ and *med23*$^{\Delta li}$ mouse livers (Fig 7B). We next examined whether the histone methyltransferase G9a affects *Ccl5* and *Cxcl10* expression and observed that, compared with the vector control, *G9a* overexpression in AML12 cells repressed *Ccl5* and *Cxcl10* mRNA expression, whereas the MED23-independent *Ccl2* mRNA level was not affected by *G9a* overexpression (Fig 7C). Together, these data suggest a mechanistic model in which the Mediator subunit MED23 may suppress RORα-directed *Ccl5* and *Cxcl10* expression through G9a-mediated H3K9 dimethylation.

## Discussion

The hepatic cellular responses to toxic drugs such as CCl$_4$ are orchestrated by multiple sophisticated events including the death of hepatocytes, infiltration of immune cells, and reactivation of quiescent HSCs, which ultimately lead to an abnormal architecture and to organ damage and dysfunction [1]. In our efforts to understand the molecular basis of liver fibrosis, we discovered that the Mediator subunit MED23 may act as a transcriptional "brake" for the expression of *Ccl5* and *Cxcl10* in hepatocytes as well as for the subsequent proinflammatory cascades,

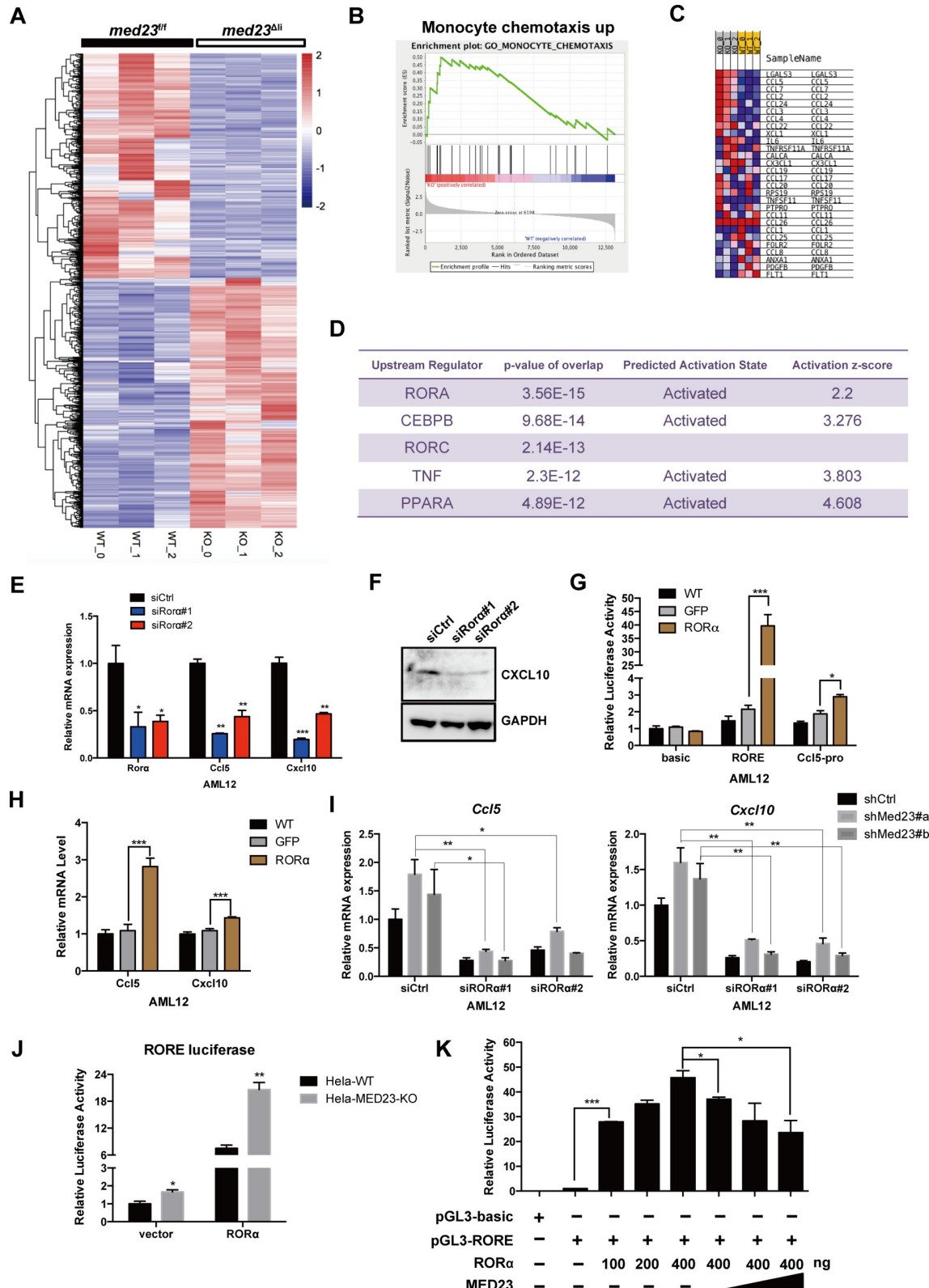

**Fig 6. Hepatocyte MED23 cooperates with RORα to regulate *Ccl5* and *Cxcl10*.** (A) Heatmap analysis of differential expression genes between *med23*^f/f and *med23*^Δli mouse livers after acute administration of CCl₄ for 24 hours. We found that 593 genes were down-regulated (>2-fold) and 660 genes were up-regulated (>2-fold) in the *med23*^Δli mice. (B-C) GSEA analysis of up-regulated

genes (>2-fold) in *med23*^Δli mice (B) and genes included in the panel B (C). (D) IPA was performed to predict the upstream regulator of up-regulated genes (>2-fold). The top five regulators are listed. (E) qRT-PCR analysis of *RORα*, *Ccl5*, and *Cxcl10* expression in AML12 cells after *RORα* knockdown (*n* = 3 per group). (F) Western blotting analysis of CXCL10 protein expression in AML12 cells after *RORα* knockdown. GAPDH was used as a loading control. (G) Effect of *RORα* overexpression on RORE and Ccl5-pro luciferase reporter activity in AML12 cells (*n* = 3 per group). (H) qRT-PCR analysis of *Ccl5* and *Cxcl10* expression in AML12 cells after *RORα* overexpression (*n* = 4 per group). (I) qRT-PCR analysis of *Ccl5* and *Cxcl10* expression in shCtrl and shMed23 AML12 cells after *RORα* knockdown. The expression was normalized to *Gapdh* (*n* = 3 per group). (J, K) Effect of MED23 on RORE-luciferase reporter activity in HeLa cell line (*n* = 3 per group). Data are presented as means ± SEM. *$P < 0.05$, **$P < 0.01$, ***$P < 0.001$. For underlying data, see S1 Data file. AML12, alpha mouse liver 12; Ccl, C-C motif chemokine ligand; CCl₄, carbon tetrachloride; CEBPB, CCAAT enhancer binding protein beta; Cxcl, C-X-C motif chemokine ligand; GAPDH, glyceraldehyde 3-phosphate dehydrogenase; GFP, green fluorescent protein; GSEA, gene set enrichment analysis; IPA, Ingenuity Pathway Analysis; KO, knockout; MED23, Mediator complex subunit 23; *med23*^Δli, liver-specific knockout of *Med23*; *med23*^f/f, *med23*-floxed; PPARα, peroxisome proliferator activated receptor alpha; qRT-PCR, quantitative real-time PCR; RORα, RAR-related orphan receptor alpha; RORE, RORα response element; shCtrl, negative control vector containing scrambled shRNA; shMed23, shRNA against Med23; siCtrl, negative control vector containing scrambled siRNA; siRORα, siRNA against RORα; TNFα, tumor necrosis factor alpha; WT, wild-type.

which curbs CCl₄-induced HSC activation and liver fibrosis (Fig 7D). In summary, the Mediator subunit MED23 is an anti-inflammatory and antifibrogenic factor in the liver.

Several types of cells can be stimulated to express and secrete cytokines/chemokines; in the liver, hepatocytes, macrophages and activated HSCs have been reported to produce and release cytokines/chemokines during the pathogenesis of liver injury and diseases [2]. The functions of these cytokines/chemokines and their receptors in the diseased liver have been established through knockout mouse models [9–11, 13], though their regulation by upstream signaling is not clear. We observed that mice with hepatocyte-specific *Med23* deletion show enhanced liver fibrosis. Consistent with this phenotype, we observed more extensive cytokine/chemokine expression in *med23*^Δli mouse livers than in control mouse livers after acute and/or chronic CCl₄ challenge. Through selective macrophage ablation via clodronate liposome injection and in vitro cell line analysis, we further discovered that among the cytokines/chemokines, CCL5 and CXCL10 appeared to be produced in a manner directly controlled by hepatic MED23, implying that hepatic MED23 functions as a negative upstream regulator of these two important chemokines. Although the hepatocytes are thought not to be the main source of these chemokines (e.g., CCL5 and CXCL10) in the liver under the normal state, some chemokines could be derived from hepatocytes, especially when hepatic homeostasis is disrupted by manipulation like deletion of *Med23* under CCl₄ treatment. We thus postulate that *Med23*-deficient hepatocytes are prone to induce excessive CCL5 and CXCL10 production at the injury site, which then creates a concentration gradient to attract additional macrophages and HSCs. The activated macrophages and HSCs communicate with each other through secreted factors to form a vicious cycle, which finally leads to augmented fibrosis. Lastly, we could not exclude the possibility at the present time that *Med23* deletion may also trigger the enlarged inflammatory responses via the additional molecular mechanisms, such as that *Med23*-deficient hepatocytes could activate other liver cells to produce the cytokines/chemokines. Overall, our study could expand the molecular network of upstream signaling in the production of chemokines associated with liver fibrosis.

To better understand how hepatocyte MED23 regulates the expression of *Ccl5* and *Cxcl10*, we investigated the upstream TFs that might function through the Mediator subunit MED23. Both the comprehensive transcriptome analysis and in vitro assays indicate that RORα, one member of the ROR family, is the partner of MED23 in the pathogenesis of liver fibrosis. RORα is reported to regulate cerebellum development [20, 35] and circadian rhythms [21, 22], as well as lipid metabolism and fat accumulation [23, 24, 36]. However, the role of RORα in impacting inflammatory responses is controversial [36–39]. RORα negatively regulated the TNFα-induced inflammatory responses in primary aortic smooth-muscle cells by inhibiting

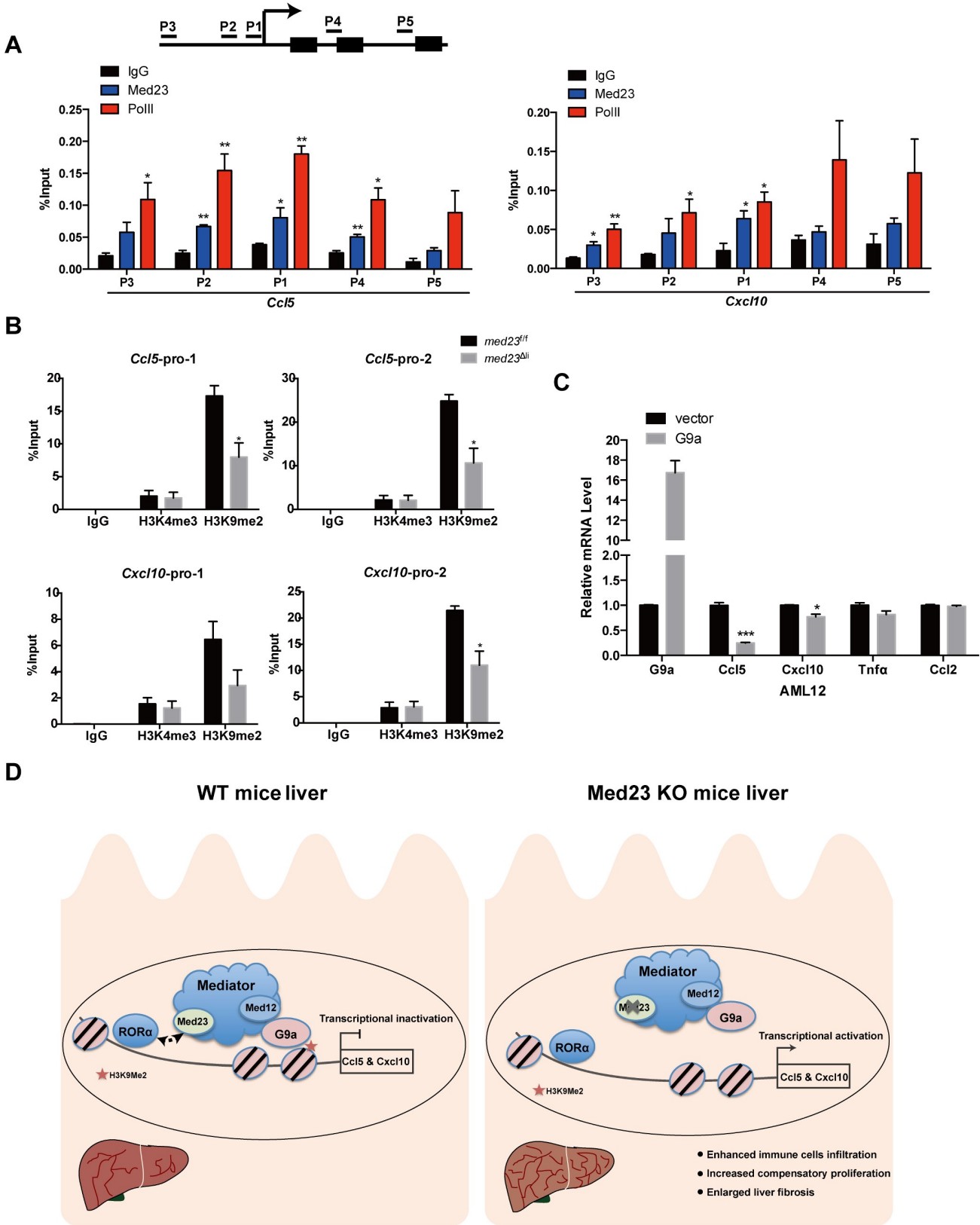

**Fig 7. *Med23* deletion causes decreased H3K9me2 occupancy among the promoter of *Ccl5* and *Cxcl10*.** (A, B) ChIP analysis of MED23 and Pol II occupancy in AML12 cells (*n* = 3 per group) (A) as well as H3K4me3 and H3K9me2 occupancy in *med23*^f/f and *med23*^Δli mouse livers after acute administration of CCl$_4$ (*med23*^f/f, *n* = 3; *med23*^Δli, *n* = 4) (B). IgG was used as a negative control. The precipitated DNA was analyzed by qRT-PCR with primers targeting the promoter regions of *Ccl5* and *Cxcl10*. The relative binding level of each factor was calculated by normalization to the input DNA. (C) qRT-PCR analysis of proinflammatory cytokines and chemokines in AML12 cells after *G9a* transient transfection (*n* = 3 per group). (D) A proposed model for the role of MED23 in hepatocyte after CCl$_4$ challenge. MED23 modulates RORα transcriptional activity possibly via G9a-mediated H3K9me2 modification to the target promoters for transcriptional repression. Data are presented as means ± SEM. *$P < 0.05$, **$P < 0.01$, ***$P < 0.001$. For underlying data, see S1 Data file. AML12, alpha mouse liver 12; Ccl, C-C motif chemokine ligand; CCl$_4$, carbon tetrachloride; ChIP, chromatin immunoprecipitation; Cxcl, C-X-C motif chemokine ligand; IgG, immunoglobulin G; KO, knockout; MED23, Mediator complex subunit 23; *med23*^Δli, liver-specific knockout of *Med23*; *med23*^f/f, *med23*-floxed; Pol II, RNA polymerase II; qRT-PCR, quantitative real-time PCR; RORα, RAR-related orphan receptor alpha; WT, wild-type.

nuclear factor kappa-light-chain-enhancer of activated B cells (NF-κB) transcriptional activity [37]. In vitro evidence supported the anti-inflammatory function of RORα in human macrophages [38]. By contrast, RORα has also been reported as a proinflammatory factor. RORα overexpression or treatment with the RORα-specific agonist increased the expression of inflammatory cytokines and increased the number of infiltrated macrophages into adipose tissue [39]. Additionally, in the adipose of RORα^sg/sg (staggerer [sg]) mice fed with an HFD, inflammatory response was significantly reduced [36]. These observations suggest that the influence of RORα on inflammatory responses is cell and tissue dependent. The potential effects of RORα on inflammatory responses in hepatocytes and liver are largely unknown. We found in this study that RORα can act as the upstream TF to specifically control *Ccl5* and *Cxcl10* expression positively in hepatocytes, thereby aggravating the immune response in liver. These observations provide a promising strategy for inhibiting the activation of inflammatory responses by RORα antagonists in liver disease development. Besides, it is worthwhile to note that MED23 may regulate *Ccl5* and *Cxcl10* expression via alternative molecules. For example, C/EBPβ, which determines differential gene activation by interacting with the Mediator MED23 subunit [40], was also predicted to regulate *Ccl5* and *Cxcl10* expression. However, we observed that knockdown of *C/EBPβ* only slightly affected the increase in *Ccl5* and *Cxcl10* expression caused by *Med23* silencing in AML12 cells, suggesting that C/EBPβ may not be the major player in this model system.

We observed that MED23 suppresses the transcriptional activity of RORα in hepatocytes through G9a-mediated H3K9 dimethylation (Fig 7D). In a previous study, hepatic RORα was showed to play a pivotal role in maintaining homeostasis of lipid metabolism by regulating PPARγ signaling through recruiting histone deacetylase 3 (HDAC3) to the target genes for transcriptional repression [24]. Interestingly, we have also previously observed that HDAC1 is among the potential interacting protein partners of MED23 [41]. In the future, it will be interesting to define the dynamic interacting protein network with which Mediator MED23 participates in the different diseases and to determine exact molecular mechanisms underlying the liver disease–related transcription regulation.

After chronic liver injury, surviving liver parenchyma cells expand to replenish the surrounding dead cells to maintain homeostasis [3]. However, in our study, although ECM deposition and liver fibrosis increased after hepatocyte-specific *Med23* deletion, we further observed increased compensatory hepatocyte proliferation in *med23*^Δli mice compared with that in control mice. These data suggest that *Med23*-deficient hepatocytes display increased regenerative capability after CCl$_4$ injury, although the detailed mechanisms remain to be further investigated. Perhaps the cross talk between hepatocytes and nonparenchymal liver cells, such as HSCs and immune cells, contributes to the induction of hepatocyte proliferation; and *Med23* deficiency in hepatocytes results in increased immune cell infiltration and HSC activation while providing additional paracrine signals (HGF, cytokines, chemokines, etc.) to

stimulate hepatic proliferation. Conceivably, modulating paracrine signaling may present interesting approaches for liver regeneration or liver cancer treatment.

In conclusion, we demonstrate for the first time that the Mediator subunit MED23 plays an important role in experimental liver fibrosis and provide new insight into the molecular mechanisms of inflammatory responses that initiate fibrotic changes upon liver injury. In the future, targeting the MED23/chemokine axis may provide intervention strategies for liver fibrosis.

## Materials and methods

### Ethics statement

All animal experiments were conducted using mice bred at and maintained in our animal facility according to the guidelines of the Institutional Animal Care and Use Committee of the Shanghai Institute of Biochemistry and Cell Biology (IACUC-PR01).

### Animals

The *med23*$^{f/f}$ mice were generated via homologous recombination [42]. *Albumin*-cre mice were a generous gift from Dr. Lijian Hui (Shanghai Institute of Biochemistry and Cell Biology, CAS, China). To generate hepatocyte-specific *med23*$^{\Delta li}$ mice, *med23*$^{f/f}$ mice were crossbred with *Albumin*-cre mice as described previously [18]. All animals were maintained under a 12-hour light/12-hour dark cycle in specific pathogen–free conditions.

### CCl₄-induced fibrosis model

The 8-week-old male mice were intraperitoneally injected with 0.6 μl/g body weight $CCl_4$ (1:4 vol/vol in sunflower oil from Sigma) every 3 days for 4 weeks to induce liver fibrosis. All mice were euthanized 72 hours after the final $CCl_4$ injection, and livers were collected for histological, cytological, biochemical, and molecular analyses. To induce acute liver injury, 8-week-old male mice were injected with $CCl_4$ as described above; all animals were euthanized 24 hours after $CCl_4$ injection.

### Cell culture

The AML12 cell line was kindly provided by Dr. Jianguo Song [43] (Shanghai Institute of Biochemistry and Cell Biology, CAS, China). The 293T and HeLa cell lines were as described previously [44, 45]. All cell lines were cultured in DMEM (HyClone, SH30243.01) supplemented with 10% fetal bovine serum (FBS), 100 U/ml penicillin, and 100 μg/ml streptomycin (Gibco, 15140122) at 37˚C with 5% $CO_2$.

### Histological analysis

HE and immunohistochemistry were performed on formalin-fixed paraffin-embedded liver sections as described previously [46]. The following antibodies were used for staining: anti-α-SMA (Abcam, ab124964), anti-PCNA (Cell Signaling Technology, 2586), and anti-Ki67 (Novocastra, NCL-Ki67p).

To detect collagen deposition, paraffin sections were stained with Sirius red dissolved in picric acid solution according to the manufacturer's recommendations.

For F4/80 immunofluorescence staining, liver cryosections were fixed in precooled acetone for 10 minutes and washed with phosphate-buffered saline (PBS) three times. After immersion in diluted normal goat serum, sections were sequentially incubated with anti-mouse F4/80 antigen PE (eBioscience, 12–4801) overnight at 4˚C. Nuclei were stained with DAPI for 5 minutes after two washes with PBS.

TUNEL assays were performed using an Apoptosis DNA Fragmentation Assay Kit (Clontech, #630107) to detect apoptotic cells. All images were visualized using a U-RFL-T microscope (Olympus, Tokyo, Japan). The positive cells or areas were counted or measured in at least five fields on each slide using ImageJ (NIH) software.

### Macrophage elimination by clodronate liposomes

A single dose of control liposomes or clodronate liposomes (10 μl/g body weight) was intravenously injected into the tail vein of 8-week-old male mice to deplete macrophages. After 24 hours, the mice were administered a $CCl_4$ injection; and after an additional 24 hours, the mice were euthanized, and F4/80 immunofluorescence staining was performed to detect the efficiency of macrophage elimination.

### RNA-seq and data analysis

The purity and integrity of the extracted RNA were confirmed using an Agilent Bioanalyzer. Libraries were prepared from 100 ng of total RNA (TruSeq v2, Illumina), and pair-ended sequencing was performed on an Illumina HiSeq 2500 using bar-coded multiplexing and a 150-bp read length, yielding a median of 34.1 M reads per sample. After quality assessment of the corresponding FASTQ files with FastQC, adapter sequences were trimmed from the reads. Read alignment and junction finding were accomplished using TopHat, and differential gene expression was analyzed with Cuffdiff using the University of California Santa Cruz (UCSC) mm10 assembly as the reference sequence. In addition, genes with an expression fold change of >|2| were selected for further analysis (accession code GSE137457).

### Construction of stable *Med23* knockdown cells

An AML12 cell line with stable *Med23* knockdown was established according to the manufacturer's recommendation (Clontech). Briefly, retroviruses were generated following the cotransfection of recombinant pSiren-RetroQ plasmids with the pCL10A1 helper plasmid into 293T cells using Lipofectamine 2000 (Invitrogen, 11668019). Culture supernatants containing retroviruses were harvested 48 hours after transfection and filtered through 0.22-μm filters. Virus-containing supernatants were mixed with 4 μg/ml Polybrene (1:100, Sigma-Aldrich) and 10 μM HEPES (pH 7.5, 1:100) and were then added to the plated cells for spin infection (2,500 rpm, 30˚C, 90 minutes). Cells were selected with puromycin (Sigma-Aldrich) 24 hours after spin infection. The targeting sequences are listed in S1 Table.

### siRNA transfection

AML12 cells were transfected with the indicated siRNAs (10 μM) using RNAiMax transfection reagent (Invitrogen, 13778150) according to the manufacturer's instructions. At 48 hours after transfection, cells were collected for protein and RNA extraction. For Fig 6I, retrovirally mediated *Med23* knockdown or the control AML12 cells were seeded into 6-well plates (150,000 cells per well). The next day, these cells were transfected with the control siRNA or RORα siRNA (10 μM) using RNAiMax transfection reagent. At 48 hours after transfection, cells were collected for protein and RNA extraction. The targeting sequences are listed in S2 Table.

### ChIP

Mouse livers (0.2 g) for ChIP assays were freshly isolated and cut into small pieces in precooled PBS supplemented with protein inhibitors. After centrifugation (450*g*, 4˚C, 5 minutes), the small pieces were cross-linked with a final concentration of 1% formaldehyde in PBS on a

shaker for 20 minutes at room temperature and neutralized by the addition of glycine to a final concentration of 0.125 M for 5 minutes. After washing with precooled PBS, the small pieces were homogenized in ChIP lysis buffer (50 mM Tris-HCl, pH 7.4; 1% SDS; and 10 mM EDTA), and the pellet was collected, followed by sonication. The following procedures were performed as described previously [18]. The antibodies used for ChIP were as follows: anti-H3K4me3 (Abcam, ab8580) and anti-H3K9me2 (Abcam, ab1220). DNA extracted from ChIP products was analyzed by qRT-PCR with TB Green Premix Ex Taq (Tli RNaseH Plus) (Takara, RR420A). The primers are listed in S3 Table.

### Dual-luciferase reporter assay

The dual-luciferase reporter assay was performed as described [47]. Briefly, HeLa and AML12 cells were seeded into a 12-well plate at $1 \times 10^5$ cells per well overnight. These cells were then transfected with a luciferase reporter plasmid and pRL-TK (Promega) along with various expression constructs, as indicated, by Lipofectamine 2000 (Invitrogen, 11668019). All wells were supplemented with control empty expression vector plasmids to keep the total amount of DNA constant. The cells were harvested and subjected to dual-luciferase reporter assays after 24–36 hours of transfection according to the manufacturer's protocol (Promega).

### Statistical analysis

All data are presented as the means ± SEM. A two-tailed unpaired Student $t$ test or Mann-Whitney test was used to determine significant differences between data sets using GraphPad Prism (version 5.0). Differences were considered statistically significant when $P \leq 0.05$.

### Supporting information

**S1 Fig. Analysis of *med23*^f/f and *med23*^Δli mice livers after feeding with MCD diet.** (A) qRT-PCR analysis of fibrosis-related factors in whole-liver extracts of *med23*^f/f and *med23*^Δli mice fed with a MCD diet (*med23*^f/f, $n = 9$; *med23*^Δli, $n = 6$). (B) Liver sections from *med23*^f/f and *med23*^Δli mice fed with a MCD diet were stained with HE and α-SMA, and representative views are shown. (C) Quantification of the α-SMA-positive area in livers of *med23*^f/f and *med23*^Δli mice (*med23*^f/f, $n = 9$; *med23*^Δli, $n = 6$). (D) The total protein was extracted from whole livers of *med23*^f/f and *med23*^Δli mice fed with a MCD diet and analyzed by western blotting using the indicated antibodies. HSP70 was used as a loading control. (E) Quantification of the α-SMA and Col1a1 levels in livers of *med23*^f/f and *med23*^Δli mice (by the gray degree value in D) and normalized to the HSP70. Error bars denote SEM from three independent experiments (*med23*^f/f, $n = 4$; *med23*^Δli, $n = 4$). Data are presented as means ± SEM. $^*P < 0.05$, $^{**}P < 0.01$, $^{***}P < 0.001$. For underlying data, see S1 Data file. α-SMA, alpha-smooth muscle actin; Col, collagen; HE, hematoxylin–eosin; HSP70, hot shock protein 70; MCD, methionine and choline-deficient; *med23*, Mediator complex subunit 23; *med23*^Δli, liver-specific knockout of *Med23*; *med23*^f/f, *med23*-floxed; qRT-PCR, quantitative real-time PCR.
(TIF)

**S2 Fig. Liver injury and liver/body weight are comparable between *med23*^f/f and *med23*^Δli mice after chronic CCl₄ administration (related to Fig 2).** (A) Serum ALT, AST, and albumin were measured in *med23*^f/f and *med23*^Δli mice ($n = 6$–$7$ per group). (B) Analysis of liver weight, body weight, and liver/body weight in *med23*^f/f and *med23*^Δli mice ($n = 7$ per group). Data are presented as means ± SEM. $^*P < 0.05$, $^{**}P < 0.01$. For underlying data, see S1 Data file. ALT, alanine aminotransferase; AST, aspartate aminotransferase; CCl₄, carbon tetrachloride; *med23*, Mediator complex subunit 23; *med23*^Δli, liver-specific knockout of *Med23*;

*med23*$^{f/f}$, *med23*-floxed.
(TIF)

**S3 Fig. Analysis of *med23*$^{f/f}$ and *med23*$^{\Delta li}$ mice livers after acute CCl$_4$ administration (related to Fig 4).** (A) Representative views of HE staining in the liver sections of *med23*$^{f/f}$ and *med23*$^{\Delta li}$ mice after acute CCl$_4$ administration for 24 hours. (B) Serum ALT and AST were measured in *med23*$^{f/f}$ and *med23*$^{\Delta li}$ mice after acute CCl$_4$ administration for 24 hours (non-treated: *med23*$^{f/f}$, $n = 7$; *med23*$^{\Delta li}$, $n = 7$; CCl$_4$ 24h: *med23*$^{f/f}$, $n = 5$; *med23*$^{\Delta li}$, $n = 4$). (C) Representative images of α-SMA and F4/80 staining in the liver sections of *med23*$^{f/f}$ and *med23*$^{\Delta li}$ mice after acute CCl$_4$ administration for 24 hours. The liver paraffin sections were stained with α-SMA antibody, and liver frozen sections were stained with F4/80 antibody (red). DAPI (blue) was used for nuclear counterstaining. (D) Quantification of F4/80-positive cells from the liver sections of *med23*$^{f/f}$ and *med23*$^{\Delta li}$ mice (*med23*$^{f/f}$, $n = 6$; *med23*$^{\Delta li}$, $n = 4$). (E-F) qRT-PCR analysis of *Acta2* (E) and *F4/80* (F) expression in whole-liver extracts of *med23*$^{f/f}$ and *med23*$^{\Delta li}$ mice. The mRNA expression was normalized to *Gapdh* (*med23*$^{f/f}$, $n = 6$; *med23*$^{\Delta li}$, $n = 5$). (G) Western blotting analysis of total protein extracted from whole livers of *med23*$^{f/f}$ and *med23*$^{\Delta li}$ mice using the indicated antibodies. GAPDH was used as a loading control. (H) qRT-PCR analysis of fibrosis-associated genes expression in whole-liver extracts of *med23*$^{f/f}$ and *med23*$^{\Delta li}$ mice (*med23*$^{f/f}$, $n = 6$; *med23*$^{\Delta li}$, $n = 5$). Data are presented as means ± SEM. $^{**}P < 0.01$. For underlying data, see S1 Data file. α-SMA, alpha-smooth muscle actin; ALT, alanine aminotransferase; AST, aspartate aminotransferase; CCl$_4$, carbon tetrachloride; GAPDH, glyceraldehyde 3-phosphate dehydrogenase; HE, hematoxylin–eosin; *med23*, Mediator complex subunit 23; *med23*$^{\Delta li}$, liver-specific knockout of *Med23*; *med23*$^{f/f}$, *med23*-floxed; qRT-PCR, quantitative real-time PCR.
(TIF)

**S4 Fig. Expressional analysis of predicted upstream regulators in liver samples of published database (related to Fig 6).** (A) The published database (GSE14323) [48] was utilized to evaluate the mRNA expression of predicted upstream regulators (normal, $n = 19$; cirrhosis, $n = 58$; HCC, $n = 38$). (B-C) qRT-PCR analysis of *RORα* (B) and *Ccl2* (C) expression in AML12 cells after RORα overexpression. The mRNA expression was normalized to *Gapdh* ($n = 4$ per group). (D) qRT-PCR analysis of *Ccl2* expression in shCtrl and shMed23 AML12 cells after *RORα* knockdown. The expression was normalized to *Gapdh* ($n = 3$ per group). $^{*}P < 0.05$, $^{**}P < 0.01$, $^{***}P < 0.001$, $^{****}P < 0.0001$. For underlying data, see S1 Data file. AML12, alpha mouse liver 12; Ccl, C-C motif chemokine ligand; *Gapdh*, glyceraldehyde 3-phosphate dehydrogenase; HCC, hepatocellular carcinoma; MED23, Mediator complex subunit 23; qRT-PCR, quantitative real-time PCR; RORα, RAR-related orphan receptor alpha; shCtrl, negative control vector containing scrambled shRNA; shMed23, shRNA against Med23.
(TIF)

**S1 Data. Numerical data used in all the figures.**
(XLSX)

**S2 Data. List of genes with log$_2$FC $\geq 1$ or $\leq 1$ and $P < 0.05$ found by RNA-seq in *med23*$^{\Delta li}$ mice livers compared with controls following 24 hours of CCl$_4$ treatment.** CCl$_4$, carbon tetrachloride; FC, fold change; *med23*, Mediator complex subunit 23; *med23*$^{\Delta li}$, liver-specific knockout of *Med23*; RNA-seq, RNA sequencing.
(XLS)

**S3 Data. The upstream regulator analysis by IPA on significantly up-regulated genes with a 2-fold increase in expression after hepatic *Med23* deletion.** IPA, Ingenuity Pathway Analysis; *Med23*, Mediator complex subunit 23.
(XLS)

**S1 Table. List of shRNA targeting sequences.** shRNA, short hairpin RNA.
(DOCX)

**S2 Table. List of siRNA targeting sequences.** siRNA, small interfering RNA.
(DOCX)

**S3 Table. List of primers used in the ChIP-qPCR analysis.** ChIP-qPCR, chromatin immuno-precipitation followed by quantitative PCR.
(DOCX)

**S4 Table. List of primers used in the qRT-PCR analysis.** qRT-PCR, quantitative real-time PCR.
(DOCX)

**S1 Text. Supporting methods.**
(PDF)

**S1 Raw Images. The raw images of western blotting displayed in this study.**
(PDF)

## Acknowledgments

We thank Drs. Li-Jian Hui, Hong-bin Ji, Gao-xiang Ge, and Li-ming Sun (Shanghai Institute of Biochemistry and Cell Biology, China) for helpful discussions and their gifts of primary antibodies and Dr. Jiu-cun Wang (Fudan University, China) for helpful advice and discussion. We also thank Dr. Sung Hee Baek (Seoul National University, Korea) for the generous gifts of plasmids.

## Author Contributions

**Conceptualization:** Zhichao Wang, Dan Cao, Gang Wang.

**Data curation:** Zhichao Wang, Dan Cao.

**Formal analysis:** Gang Wang.

**Funding acquisition:** Gang Wang.

**Investigation:** Zhichao Wang, Dan Cao, Gang Wang.

**Methodology:** Chonghui Li, Lihua Min.

**Project administration:** Zhichao Wang, Gang Wang.

**Resources:** Zhichao Wang, Dan Cao, Chonghui Li, Lihua Min.

**Software:** Chonghui Li.

**Supervision:** Gang Wang.

**Visualization:** Zhichao Wang, Dan Cao.

**Writing – original draft:** Zhichao Wang, Dan Cao, Gang Wang.

**Writing – review & editing:** Gang Wang.

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
