## [Editor Report · Decision Letter 0]

7 Jun 2019

Dear Dr WANG, 

Thank you for submitting your manuscript entitled "The Mediator subunit MED23 regulates inflammatory responses and liver fibrosis" for consideration as a Research Article by PLOS Biology.

Your manuscript has now been evaluated by the PLOS Biology editorial staff as well as by an academic editor with relevant expertise and I am writing to let you know that we would like to send your submission out for external peer review.

Please re-submit your manuscript within two working days, ie. by Jun 09 2019 11:59PM.

Kind regards,

Di Jiang

PLOS Biology

---

## [Decision Letter · Decision Letter 1]

17 Jul 2019

Dear Dr WANG,

Thank you very much for submitting your manuscript "The Mediator subunit MED23 regulates inflammatory responses and liver fibrosis" for consideration as a Research Article at PLOS Biology. Your manuscript has been evaluated by the PLOS Biology editors, an Academic Editor with relevant expertise, and by three independent reviewers.

In light of the reviews (below), we will not be able to accept the current version of the manuscript, but we would welcome resubmission of a revised version that takes into account the reviewers' comments. Our Academic Editor advises that you should include another injury model, primary hepatocyte data and data to show that this is relevant to human liver disease. It is not necessary to provide evidence that this is pathway is dysregulated in human disease, but you are encouraged to provide it if it is already available as it will increase the impact of the study. We cannot make any decision about publication until we have seen the revised manuscript and your response to the reviewers' comments. Your revised manuscript is also likely to be sent for further evaluation by the reviewers.

Your revisions should address the specific points made by each reviewer. Please submit a file detailing your responses to the editorial requests and a point-by-point response to all of the reviewers' comments that indicates the changes you have made to the manuscript. In addition to a clean copy of the manuscript, please upload a 'track-changes' version of your manuscript that specifies the edits made. This should be uploaded as a "Related" file type. You should also cite any additional relevant literature that has been published since the original submission and mention any additional citations in your response. 

Before you revise your manuscript, please review the following PLOS policy and formatting requirements checklist PDF: http://journals.plos.org/plosbiology/s/file?id=9411/plos-biology-formatting-checklist.pdf. It is helpful if you format your revision according to our requirements - should your paper subsequently be accepted, this will save time at the acceptance stage.

Please note that as a condition of publication PLOS' data policy (http://journals.plos.org/plosbiology/s/data-availability) requires that you make available all data used to draw the conclusions arrived at in your manuscript. If you have not already done so, you must include any data used in your manuscript either in appropriate repositories, within the body of the manuscript, or as supporting information (N.B. this includes any numerical values that were used to generate graphs, histograms etc.). For an example see here: http://www.plosbiology.org/article/info%3Adoi%2F10.1371%2Fjournal.pbio.1001908#s5.

For manuscripts submitted on or after 1st July 2019, we require the original, uncropped and minimally adjusted images supporting all blot and gel results reported in an article's figures or Supporting Information files. We will require these files before a manuscript can be accepted so please prepare them now, if you have not already uploaded them. Please carefully read our guidelines for how to prepare and upload this data: https://journals.plos.org/plosbiology/s/figures#loc-blot-and-gel-reporting-requirements.

Upon resubmission, the editors will assess your revision and if the editors and Academic Editor feel that the revised manuscript remains appropriate for the journal, we will send the manuscript for re-review. We aim to consult the same Academic Editor and reviewers for revised manuscripts but may consult others if needed.

We expect to receive your revised manuscript within two months. Please email us (plosbiology@plos.org) to discuss this if you have any questions or concerns, or would like to request an extension. At this stage, your manuscript remains formally under active consideration at our journal; please notify us by email if you do not wish to submit a revision and instead wish to pursue publication elsewhere, so that we may end consideration of the manuscript at PLOS Biology.

When you are ready to submit a revised version of your manuscript, please go to https://www.editorialmanager.com/pbiology/ and log in as an Author. Click the link labelled 'Submissions Needing Revision' where you will find your submission record. 

Sincerely,

Di Jiang

PLOS Biology

Reviewer remarks:

Reviewer #1: Wang et al, investigated the role of the MED23 subunit of the transcriptional Mediator complex in CCl4-induced liver injury. Mice with hepatic Med23 deletion exhibited aggravated CCl4-induced

liver fibrosis, enhanced inflammation and increased hepatocyte regeneration. The authors suggest that this phenotype is mediated by MED23-induced suppresses of orphan nuclear receptor RORα, which regulates the activation of liver fibrosis-related chemokines CCL5 and CXCL10.

Major points

1. Half of the paper is focusing on the CCl4 responses in MED23-deleted mice that show increased HSC activation, fibrosis, inflammation and hepatocyte proliferation while hepatocyte apoptosis is reduced (Fig.1-4, half of the figures). While this data is relevant, most of this data is descriptive and may be related to the same mechanisms, meaning that it is likely looking at the same observation from different angles.

2. The paper relies on a single model and the rationale for studying MED23 is not well explained. The authors should employ more liver injury models to exclude that their findings are specific to CCl4, a standardized model for liver fibrosis research but without much human relevance. Moreover, they should analyze MED23 in patients. It is not clear at all whether MED23 activity is altered in human fibrosis and contributes to disease progression.

3. The authors do not sufficiently investigate whether MED23 may simply regulate responses to CCl4 and liver injury. The authors show ALT levels in the chronic CCl4 model, where liver injury is typically low towards the end of experiments and where differences may be difficult to reveal. There is no data on serum ALT in the acute injury models and not investigation of key factors that regulate CCl4 metabolization and liver injury such as Cyp expression/activity. Moreover, the assessment of apoptosis by TUNEL assay and cleaved caspase is not highly relevant as most of CCl4-induced cell death is necrosis. In light of these, it would also have been advantageous to use additional models that do not rely on biactivation of a toxin and/or mostly necrotic cell death (see comments above).

4. The clodronate and in vitro experiments do not conclusively prove a role for MED23 via the regulation of hepatocyte Ccl5 and Cxcl10. The authors should have analyzed this in primary hepatocytes and possbily used single cell RNA-seq to compare Ccl5 and Cxcl10 expression between MED23 and floxed controls as well as between hepatocytes and other cell types in the MED23-deleted mice - this way they could have strengthened their hypothesis that hepatocytes represent the main Ccl5- and Cxcl10-secreting cell type in mice with MED23 in hepatocytes. Finally, it would have been important to show the role of Ccl5 and/or Cxcl10 by deleting these in hepatocytes in vivo - which can nowadays be achieved by hepatocyte-specific silencing or deletion methods without lengthy breedings.

5. The authors found that Hgf, was higher in MED23-deleted livers than in control livers. It is not clear why this is the case as HGF is normally expressed by stellate cells and endothelial cells. As Hgf is an important modulator of liver injury and regeneration, this could also represent a major mechanism by which MED23 regulates responses to liver injury, rather than the postulated Ccl5 and Cxcl10 pathway in hepatocytes.

Minor points

1. The authors state “the expression of Desmin, another marker of HSCs activation, was also higher in med23Δli mouse livers than in med23f/f mouse livers”. Desmin is not a marker of activation - it is a marker of HSC and an increase may reflect an increased number of HSC in injured livers.

Reviewer #2: Liver fibrosis is a wound-healing response that is a consequence of chronic liver diseases. Despite the widespread of this disease, the underlying molecular mechanisms remain largely unknown. Development of novel effective therapies for liver fibrosis requires identifying effective molecular targets. The focus of this research is the study of the molecular mechanism underlying liver fibrosis that is a relevant and important issue.

The manuscript “The Mediator subunit MED23 regulates inflammatory responses and liver fibrosis” is dedicated to studying of the role MED23 subunit of the Mediator complex in the development of experimental liver fibrosis. The authors showed that mice with hepatic Med23 deletion exhibited aggravated CCl₄-induced liver fibrosis. The manuscript provides evidence that MED23 negatively regulates the expression of the hepatic chemokines CCL5 and CXCL10 via suppressing the activity of the retinoid-related orphan nuclear receptor, RORα.

In summary, this manuscript is of value for the further development of our knowledge of the molecular mechanism underlying liver fibrosis and will help the development of more effective treatment options. I would recommend the manuscript for publication, but I have the following comments.

CRITIQUE

My main problem with this manuscript is that the authors made several deficiencies, which cast doubt on the conclusions and correctness of the model proposed.

1. Necessary to give data about the number of mice in all experiments.

2. Fig 6F. Necessary to show Western-Blot for proving the effectivity of the knocked down Rora in protein level.

3. Fig 6I. Necessary to describe this experiment with all details in Materials and Methods. I have doubts about the conclusions made by the authors. Controls have significant differences. It should be discussed in the manuscript with an explanation. In the current moment, conclusion about a functional interaction between RORα and MED23 in controlling Ccl5 and Cxcl10 expression needs additional experiments.

4. Fig 6J. The dual luciferase reporter assay should be described in Materials and Methods.

5. Fig 6K. This experiment should be described with details in the manuscript.

6. Fig 7A. Necessary to show a base level of signal for ChIP on non-promoter site of the genome.

7. I would like to ask the authors to provide more discussion in the text. I highly recommend expanding the Introduction part and give more information about the factors used in the work.

8. In this manuscript, a significant part of the data is the results of the qRT-PCR. I would like to ask the authors to pay more intention for presenting this type of data. Nowadays good practice to use two or three genes for normalization especially if the differences between experiment and control less than 2-folds.

Reviewer #3: The authors study the role of Med23 in experimental models of liver fibrosis (carbon tetrachloride) in conditional knock out mice using Albumin-CRE, Med23 flox/flox mice. The authors find that deletion of Med23 in hepatocytes and cholangiocytes leads to increased liver fibrosis, decreased apoptosis but increased hepatocyte proliferative activity and increased inflammatory activity, though the characterization of inflammatory cell composition remains a bit superficial. In well designed in vitro studies using murine hepatocyte cell line AML12 and other models, the authors show that Med23 cooperates with ROR� in negatively regulating cytokines CCL5 and CXCL10. The authors show that Med23 deletion causes decreased H3K9me2 occupancy among the promoter of CCL5 and CXLC10 thus identifying novel mechanism for Med23. 

Though the authors ultimately do not show the biological relevance of Med23-CCL5/CXLC10 axis in liver fibrosis and inflammation e.g. by showing abrogation of increased liver fibrosis in double knock out mice of both Med23 and CCL5/CXCL10 or data on Med23 expression levels in human liver datasets from public databases (Suppl. Fig. 3), the shown data are of very good quality and experiments are well designed. The conclusions are sound and the manuscript well written with the topic within the journals scope, I thus recommend the acceptance of the manuscript for publication.

Minor comments: 

additional information regarding the catalog numbers of siRNA / plasmids used to knock down targets would be informative for readers interested in replicating the studies.

---

## [Decision Letter · Decision Letter 2]

29 Oct 2019

Dear Dr WANG,

Thank you for submitting your revised Research Article entitled "Mediator MED23 regulates inflammatory responses and liver fibrosis" for publication in PLOS Biology. I have now obtained advice from two of the original reviewers and have discussed their comments with the Academic Editor. 

The reviews are included below. We recognise the concerns that continue to be raised by reviewer 1, but after discussing them with the academic editor, we think that these can be satisfactorily addressed by ensuring that these potential limitations are clearly flagged for the reader. We will probably accept this manuscript for publication, assuming that you will modify the manuscript to address the remaining points raised by reviewer 1. Please also make sure to address the data and other policy-related requests noted at the end of this email.

We expect to receive your revised manuscript within two weeks. Your revisions should address the specific points made by reviewer 1. In addition to the remaining revisions and before we will be able to formally accept your manuscript and consider it "in press", we also need to ensure that your article conforms to our guidelines. A member of our team will be in touch shortly with a set of requests. As we can't proceed until these requirements are met, your swift response will help prevent delays to publication.

Sincerely,

Di Jiang

PLOS Biology

ETHICS STATEMENT:

Please add an Ethics Statement subsection at the beginning of the Methods section. The Ethics Statements in the submission form and Methods section of your manuscript should match verbatim. 

DATA POLICY:

Reviewer remarks:

Reviewer #1: While the authors did additional work including a second model, most of my concerns remain.

1. The second model (MCD model) was not included in the actual paper and mechanisms seem to be different from the CCl4 model and not be mediated by the proposed inflammatory mechanism. Moreover, the MCD model is nowadays viewed as inadequate model of NASH fibrosis and other dietary models are preferred (but it is adequate for purely mechanistic studies on fibrosis)

2. The authors have not convinced this reviewer that the main role of MED23 is not related to protection from injury. This appears to be THE driver of the phenotype - everything else is likely secondary unless proven otherwise by functional experiments (see also point below). 

3. This reviewer is not convinced that hepatocyte-secreted CCl5 and CXCL10 are the main drivers of fibrogenesis in this model. First of all, hepatocytes are not thought to be the main source of these chemokines in the liver. Second of all, hepatocyte isolations from CCl4-treated livers are inherently contaminated with inflammatory cells. The suggestion of using single cell RNA-seq as a platform that can avoid these problems was not taken up by the authors. In the absence of single cell RNA-seq analysis, the authors would need to provide evicence for this unlikely hypothesis, e.g. by hepatocyte-specific silencing of CCl5 and CXCL10 (which can nowadays be achieved without crossing mice, e.g. by AAV8- or GalNAc-mediated silencing).

Reviewer #2: The manuscript “Mediator MED23 regulates inflammatory responses and liver fibrosis” was modified, taking into account all the comments. The authors have conducted additional experiments to cement their findings further, as suggested by editors and reviewers. The text of the manuscript was corrected and significantly improved. I recommend this manuscript to be accepted for publication in its current form.

---

## [Editor Report · Decision Letter 3]

15 Nov 2019

Dear Dr WANG,

On behalf of my colleagues and the Academic Editor, Luke A Noon, I am pleased to inform you that we will be delighted to publish your Research Article in PLOS Biology. 

Early Version

PRESS 

Kind regards,

Hannah Harwood

Publication Assistant, 

PLOS Biology

on behalf of

Di Jiang,

Associate Editor

PLOS Biology